# The Cluster spacecraft's view of the motion of the high-latitude magnetopause

Niklas Grimmich[1], Ferdinand Plaschke[1], Benjamin Grison[2], Fabio Prencipe[1], Christophe Philippe Escoubet[3], Martin Owain Archer[4], Ovidiu Dragos Constantinescu[1,5], Stein Haaland[6,7,8], Rumi Nakamura[9], David Gary Sibeck[10], Fabien Darrouzet[11], Mykhaylo Hayosh[2], and Romain Maggiolo[11]

[1]Institut für Geophysik und Extraterrestrische Physik, Technische Universität Braunschweig, Braunschweig, Germany
[2]Department of Space Physics, Institute of Atmospheric Physics Czech Academy of Sciences, Praha, Czech Republic
[3]ESA European Space Research and Technology Centre, Noordwijk, Netherlands
[4]Department of Physics, Imperial College London, London, UK
[5]Institute for Space Sciences, Bucharest, Romania
[6]Birkeland Centre for Space Science, University of Bergen, Bergen, Norway
[7]Max-Planck-Institut für Sonnensystemforschung, Göttingen, Germany
[8]The University Center in Svalbard, Longyearbyen, Norway
[9]Space Research Institute, Austrian Academy of Sciences, Graz, Austria
[10]NASA Goddard Space Flight Center, Greenbelt, Maryland, USA
[11]Royal Belgian Institute for Space Aeronomy, Brussels, Belgium

**Correspondence:** Niklas Grimmich (n.grimmich@tu-braunschweig.de)

**Abstract.** The magnetopause is the boundary between the interplanetary magnetic field and the terrestrial magnetic field. It is influenced by different solar wind conditions, which lead to a change in the shape and location of the magnetopause. The interaction between the solar wind and the magnetosphere can be studied from in-situ spacecraft observations. Many studies focus on the equatorial plane, as this is where recent spacecraft constellations such as THEMIS or MMS operate. However, to fully capture the interaction, it is important to study the high latitude regions as well. Since the Cluster spacecraft operate in a highly elliptical polar orbit, the spacecraft often pass through the magnetopause at high latitudes. This allow us to collect a dataset of high-latitude magnetopause crossings and study magnetopause motion in this region, as well as deviations from established magnetopause models. We use multi-spacecraft analysis tools to investigate the direction of magnetopause motion in the high latitudes and compare the occurrence of crossings at different locations with the result in the equatorial plane. We find that the high-latitude magnetopause motion is generally consistent with previously reported values and seems to be more often associated with a closed magnetopause boundary. We show that on average the magnetopause moves faster inwards than outwards. Furthermore, the occurrence of magnetopause positions beyond those predicted by the Shue et al. (1998) model at high latitudes is found to be caused by the similar solar wind parameters as in the equatorial plane. Finally, we highlight the importance of the dipole tilt angle at high latitudes. Our results may be useful for the interpretation of plasma measurements from the upcoming SMILE mission (Branduardi-Raymont et al., 2018), as this spacecraft will also fly frequently through the high-latitude magnetopause.

# 1 Introduction

The Earth's magnetic field is an obstacle to the super-magnetosonic solar wind, which is deflected around the magnetosphere. The magnetopause (MP) is the boundary between the region of the redirected flow, called the magnetosheath, and the terrestrial magnetic field. To first order, this boundary is defined by a balance between the dynamic, plasma (thermal) and magnetic pressures (from the draped field lines) on the magnetosheath side and the magnetic pressure on the magnetospheric side (e.g., Shue and Chao, 2013). The MP is considered (especially in popular models such as the Shue et al., 1997) to be a closed boundary with a smooth surface and a rigid shape separating the IMF from the Earth's magnetic field. However, near the so-called cusp, the Earth's magnetic field lines can connect with the IMF field lines and be considered open, allowing energy to be transported into the inner magnetosphere. This creates a funnel-like structure in the magnetic field configuration, separating field lines that bend sunwards from field lines that bend tailwards, causing the MP surface to indent as pressure equilibrium is reached at points closer to Earth (e.g., Pitout and Bogdanova, 2021). Furthermore, dynamical changes in the solar wind pressure or in the interplanetary magnetic field (IMF) lead to continuous variation in the magnetopause location and shape (e.g., Sibeck et al., 1991, 2000; Shue et al., 1997; Plaschke et al., 2009a, c; Dušík et al., 2010).

Spacecraft constellations like Cluster (Escoubet et al., 2001, 2021), THEMIS (Angelopoulos, 2008) and MMS (Burch et al., 2016) often observe the moving and undulated MP in response to these changes as it passes over the satellites. Hence, identifying magnetopause crossings (MPCs) in the data is necessary to study the dynamics of the MP and the interaction of the solar wind with the magnetosphere. Due to its highly elliptical polar orbit of $4 \times 19.6\ R_{\mathrm{E}}$, the Cluster spacecraft is well suited to study the MP at high latitudes and at the cusp, which were some of the main objectives of this mission (Escoubet et al., 2001). THEMIS and MMS, on the other hand, are more focused on studies in the equatorial plane.

There have been many studies on the identification of MPCs for different spacecraft missions leading to construction of multiple datasets (e.g., Staples et al., 2020; Nguyen et al., 2022; Grimmich et al., 2023a, as some of the most recent studies). Most of these studies are focused on crossings in the equatorial plane and only a few datasets include the high latitude crossings (e.g., Boardsen et al., 2000; Panov et al., 2008; Petrinec et al., 2023). The study of these datasets and the fitting or comparing of MP models (e.g. Fairfield, 1971; Sibeck et al., 1991; Shue et al., 1997; Boardsen et al., 2000; Chao et al., 2002; Lin et al., 2010; Liu et al., 2015) to these datasets has already uncovered much of the basic behaviour of MP motion.

Under strong southward IMF conditions, magnetic reconnection (Levy et al., 1964; Paschmann et al., 1979, 2013) occurs, and according to Dorville et al. (2014) the MP can be best described as a composition of a compressional boundary and a rotational discontinuity (RD) under these conditions. Magnetic reconnection leads to inward motion of the MP (even up to the geosynchronous orbit) due to dayside flux erosion (Aubry et al., 1970; Sibeck et al., 1991; Shue et al., 1997, 1998; Kim et al., 2024) or undulation of the MP surface due to passage of a transient flux transfer event (Elphic, 1995; Fear et al., 2017). Magnetospheric expansions and outward motion of the MP are often found when the IMF is quasi-radial and the IMF cone angle $\vartheta_{\mathrm{cone}}$ between the Earth-Sun line and the IMF vector is less than $30°$ (Fairfield et al., 1990; Merka et al., 2003; Suvorova et al., 2010; Dušík et al., 2010; Samsonov et al., 2012; Park et al., 2016; Grygorov et al., 2017). Furthermore, the development of MP surface waves (Plaschke et al., 2009b; Archer et al., 2019), the impact of foreshock transients (Sibeck et al., 1999;

Jacobsen et al., 2009; Turner et al., 2011; Archer et al., 2015; Zhang et al., 2022; Grimmich et al., 2024b) and magnetosheath jets (Plaschke et al., 2018; Escoubet et al., 2020) and the occurrence of Kelvin-Helmholtz instabilities (Kavosi and Raeder, 2015; Michael et al., 2021) are other processes that contribute to the undulation and constant motion of the MP.

Besides the influence of solar wind dynamic pressure, IMF strength and IMF orientation on MP location and shape, the dipole tilt angle $\psi$, which describes the orientation of the Earth's dipole axis with respect to the Earth-Sun line, is also reported to strongly influence MP location, especially at higher latitudes (e.g. Boardsen et al., 2000; Lin et al., 2010; Liu et al., 2012). Furthermore, for the equatorial plane, the study by Grimmich et al. (2023a) shows that for the occurrence of large displacements of the MP from its nominal position, possibly associated with large amplitude MP motion, solar wind parameters such as the solar wind velocity or the Alfvén Mach number are important. However, there is no large-scale study showing similar effects at higher latitudes.

In order to characterize the motion of the MP, the normal (flapping) velocity $v_{MP}$ of the MP is often used. Previous studies have found that the average MP velocity in the subsolar region is about $40 \mathrm{~kms}^{-1}$ (Plaschke et al., 2009a), while on the flanks the MP velocity distribution shows an asymmetry with an average of $64 \mathrm{~kms}^{-1}$ on the dawn flank and $42 \mathrm{~kms}^{-1}$ on the dusk flank (Haaland et al., 2014). Furthermore, Panov et al. (2008) found that dayside MP motion is about 30 % slower in high latitudes than in low latitudes. However, the results from Plaschke et al. (2009a) showed agreement between equatorial mean MP motion and high latitude values from Panov et al. (2008), challenging the studies' results. Unfortunately, all these studies only give absolute values for $v_{MP}$ and not specifically analysed the direction of motion (i.e., the sign of $v_{MP}$), which plays an important role in the dynamics of the magnetosphere.

In general, as Haaland et al. (2021) has pointed out, the Cluster spacecraft data, particularly for the dayside high latitude regions, are under-utilised in studies of the magnetosphere and the MP. To our knowledge, Panov et al. (2008) did one of the few dedicated statistical investigations regarding the high latitude MP with a limited dataset of roughly 50 "proper" crossings from the Cluster data. Further analysis of the high-latitude MP motion and response to solar wind influences is therefore needed. In order to do this, it is necessary to have a larger dataset that covers the MP in the high latitude regions and on the dayside.

Therefore, we present here in the following one of the largest MPC databases of Cluster data, including the years 2001 to 2020, adapting the identification method introduced in Grimmich et al. (2023a). After validating this huge dataset with independent data (section 3), we investigate the MP motion in the high latitude regions (section 4). In addition, we determine whether certain solar wind parameters favour the occurrence of large undulations and displacements from the nominal MP position (section 5), before discussing our results (section 6).

## 2 Magnetopause crossing identification

In order to construct a Cluster MPC database similar to the THEMIS database by Grimmich et al. (2023b), we utilize a slightly modified version of the machine learning detection method introduced in Grimmich et al. (2023a). As a detailed description on the detection method is given in Grimmich et al. (2023a), we only indicate important changes and otherwise refer to the publication.

For the identification of MPCs, we use the magnetic field data from the Fluxgate Magnetometer (FGM, Balogh et al., 1997, 2001), and particle data and moments from the Cluster Ion Spectrometry Hot Ion Analyser (CIS-HIA, Rème et al., 1997, 2001). The magnetic field and plasma moments data are used in spin-averaged resolution with cadences of about 4 s during pre-processing and resampled by taking a data point every 60 s for the identification. However, we can only use data from both instruments between 2001 and 2020 for C1 (Rumba) and between 2001 and 2009 for C3 (Samba) due to the limited availability of HIA data (see Laakso et al., 2010; Dandouras et al., 2010, for details).

In contrast to the THEMIS spacecraft, which mainly operate in the equatorial plane (Angelopoulos, 2008), Cluster spent most the first years of its mission in polar orbits, studying the high-latitude magnetospheric regions. These regions are characterized by slightly different plasma regimes in comparison to the equatorial plane (e.g. Panov et al., 2008). A direct application of the trained model of Grimmich et al. (2023a) would certainly lead to more mispredictions of magnetospheric and non-magnetospheric regions. Thus, we used the same basic algorithm as Grimmich et al. (2023a), but trained this Random Forest Classifier (RFC) independently on the Cluster data, for which we have built a new training dataset.

First, we neglect time intervals where the HIA quality flag indicates insufficient data or the instrument is switched off (for details on the HIA quality flags and the data availability see Dandouras et al., 2010). We interpolate small data gaps of a few minutes (up to a maximum of 10 minutes) in the magnetic field and plasma moments data intervals if applicable, and we also interpolate data points where the quality flag only indicates a few insufficient data points. Please note that the HIA quality flag is no longer available after 01 January 2015. Therefore, data collected after this date may contain insufficient data points influencing our results. As all the crossings found after this date have been manually checked, we have included the data in order to obtain the largest possible temporal coverage.

To properly train the RFC, we select 78 random intervals from the C3 data, which should contain crossings according to the MP dataset by Petrinec et al. (2023). The data in these intervals are resampled at a cadence of 60 s and manually labelled, focusing on the energy flux density, ion density and magnetic field data for identification. We use a steep anti-correlated jump in the ion density and magnetic field data as a proxy for the spacecraft transition from one region to the other, and focus on the appearance and width of the energy flux density around 3 keV, as the narrow distribution around 3 keV often indicates solar wind measurements, and broader distributions around 3 keV indicate magnetosheath plasma measurements.

The manual labelling results in a rather uneven distribution between magnetospheric and non-magnetospheric labelled data points, which can lead to problems in the training process of the RFC. For example, if the training set contains drastically more magnetospheric data points, the trained algorithm will tend to always predict one point to be magnetospheric, leading to more false identifications. However we can label additional data points to balance our training set. Since typical values for the plasma outside the magnetosphere in the magnetosheath and the solar wind are 7 to 30 cm$-3$ for the density, around 100 km/s for the velocity and between 5 and 60 nT for the magnetic field strength (e.g. Baumjohann and Treumann, 1997; Soucek and Escoubet, 2012; Raymer, 2018), we can define thresholds to label additional data points. In Fig. 1 we show 2D histograms of the collected Cluster data for magnetic field, ion velocity and ion density. It can be seen that the data can be categorised into at least two distinct groups. Bearing in mind the typical values for data points outside the magnetosphere, we defined the

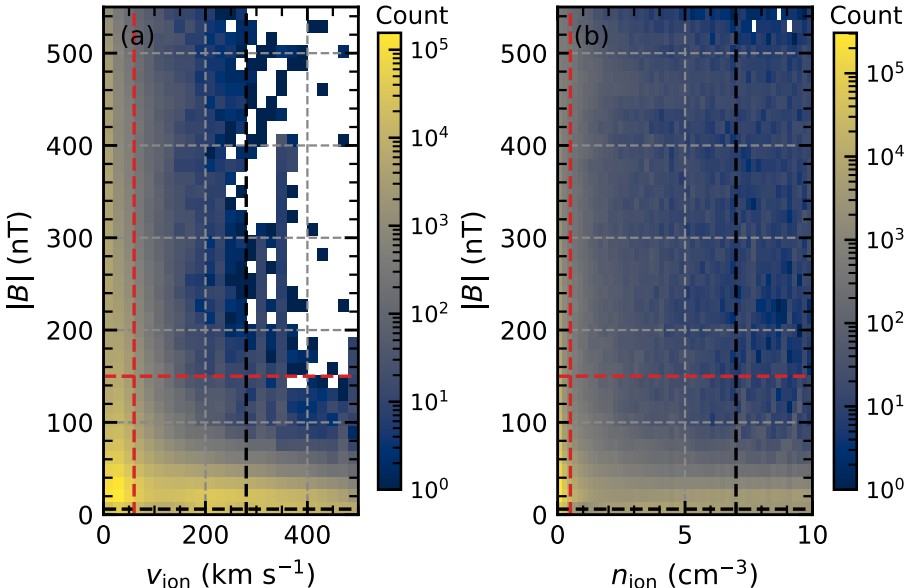

**Figure 1.** 2D distribution of the cluster data collected for MPC identification. Panel (a) shows the distribution of the measured magnetic field strength $|B|$ over the ion velocity $v_{ion}$, while panel (b) shows the distribution of $|B|$ over the ion density $n_{ion}$. Distinct groups can be identified and assigned to data inside or outside the magnetosphere, and we mark thresholds with red and black dashed lines as selection criteria for data points that are certain to belong to one of the groups.

thresholds, shown as red and black dashed lines in Fig. 1, in such a way that we could be sure to include only data points far from any group/plasma boundaries.

In detail, we used the following assumptions and thresholds to label additional data points:

1. We assume that the spacecraft is outside the magnetosphere when HIA is operating in solar wind mode, which is activated based on the modelled location of the bow shock following Howe and Binsack (1972). Thus, we label all data points in this operating mode as being outside of the magnetosphere.

2. We assume that high magnetic field magnitudes are only reached inside the magnetosphere. Thus, we label all data points
as inside the magnetosphere, if the field magnitude $B$ is greater than 450 nT.

3. Low magnetic field magnitudes ($B < 6$ nT), high ion velocities ($v > 280$ kms$^{-1}$) and high ion densities ($n_{ion} > 7$ cm$^{-3}$) are more likely observed outside the magnetosphere. Thus, if all three conditions are fulfilled we label the associated points as outside the magnetosphere.

4. High magnetic field magnitudes ($B > 150$ nT), low ion velocities ($v < 60$ kms$^{-1}$) and low ion densities ($n_{ion} < 0.5$
130    cm$^{-3}$) are more likely observed inside the magnetosphere. Thus, if all three conditions are fulfilled we label the associated points as inside the magnetosphere.

A portion of these threshold labelled data is added to the training data in order to have an even distribution of data points inside and outside the magnetosphere for training.

We train the RFC with the same parameters as Grimmich et al. (2023a) on our new training data, i.e. an input consisting of the magnetic field ($B_x$, $B_y$, $B_z$, $|B|$), the ion velocity ($v_x$, $v_y$, $v_z$, $|v|$), the ion density $n_{\text{ion}}$, the ion temperature $T_{\text{ion}}$, the flux index $F_{\text{idx}}(t)$ as a proxy for the omnidirectional ion energy flux between $10^2$ eV and $10^4$ eV. For this input, the algorithm calculates in which region the data was most likely measured and gives a probability value $p_{\text{RFC}}(t)$ for the certainty of its calculations.

Validation of the training process using unseen data points for the trained RFC gives a precision score of 0.998, i.e., 99.8 % of the predictions giving the label inside the magnetosphere are correct. The trained RFC is then used to classify whether or not Cluster is observing data from inside the magnetosphere and infer crossings as changes in the label prediction of the classifier. In addition to the quality flag from HIA (if applicable), the quality for each crossing is primarily indicated by the crossing probability derived from the RFC prediction. The crossing probability indicates how accurately the RFC can determine the labels of the data points around the crossing, thus providing a quantification of the ambiguity of the crossing. We calculate this crossing probability following Grimmich et al. (2023a) by using the prediction probability $p_{\text{RFC}}(t)$ given by the RFC and the weighted average of this probability of the 2 points before and after the jump in the label predictions:

$$p_{\text{MPC}}(t_0) = \frac{1}{3} \left[ p_{\text{RF}}(t_0 - 60\,\text{s}) + 0.5 p_{\text{RF}}(t_0) + 0.5 p_{\text{RF}}(t_0 + 60\,\text{s}) + p_{\text{RF}}(t_0 + 120\,\text{s}) \right]. \tag{1}$$

Turning the HIA instrument off and on results in signatures identifiable as MPCs, especially in the C1 data after November 2012, when the instrument was only on for selected 1 h intervals. Hence, we manually remove some of the misidentified crossings after visual inspection.

Note that unusual MPCs occurring near or outside the modelled bow shock location may be discarded in our identification method due to assumption (1). Although we know that such unusual events can occur (e.g., Grimmich et al., 2024b, reported on such an event), the different measurement mechanism of HIA in solar wind mode makes it necessary to exclude these measurements from the identification. The energy flux density, one of the main parameters used in the identification, does not clearly show the hot ion distribution around 3 keV in the solar wind mode. This distribution is often used to identify solar wind and magnetosheath regions, and is also used in our RFC. Thus, solar wind measurements (in solar wind mode) can be confused with magnetospheric measurements by the RFC, leading to unwanted false identifications.

We find 22,357 MPCs in C1 and 15,965 MPCs in C3, giving a total of 38,322 identified MPCs. In Fig. 2 (a) we plot the distribution of the observed crossing position in aberrated geocentric solar ecliptic (aGSE) coordinates in a ($x$, $R$) plane, where $R = \text{sgn}(y)\sqrt{y^2 + z^2}$ (similar to the Figure 2 shown in Mieth et al., 2019, and only used in our study for Fig. 2). The other panels of Fig. 2 show in (b) the distribution of the crossing position in latitude over longitude and in (c) a histogram of the crossing probability.

We use an average aberration angle of $\varphi \sim 4.3°$, resulting from the Earth's orbital velocity of 30 $\text{kms}^{-1}$ around the Sun and an average solar wind velocity of 400 $\text{kms}^{-1}$, to rotate the normal geocentric solar ecliptic coordinate system about the $z$-axis

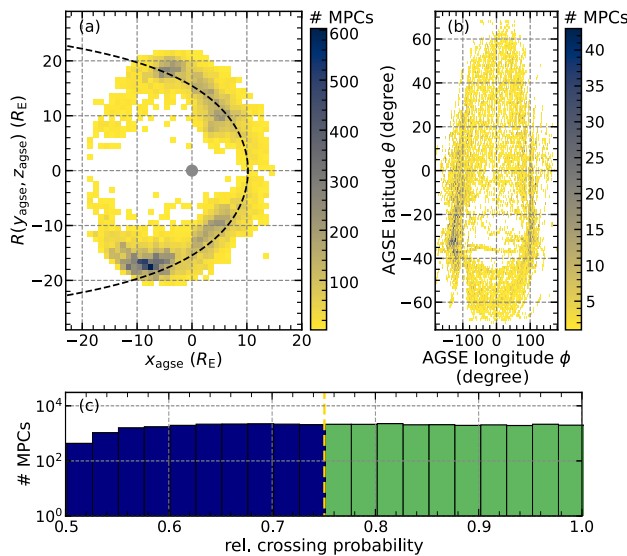

**Figure 2.** Distribution and features of the Cluster MPC database. Panel a) shows the spatial 2D distribution of the identified MPCs with a bin size of 1x1 $R_E$ plotted on the $(x_{\text{agse}}, R)$ plane, with $R = \text{sgn}(y_{\text{agse}})\sqrt{y_{\text{agse}}^2 + z_{\text{agse}}^2}$. The dashed black line shows the Shue et al. (1998) MP model for $B_z = -1$ nT and $p_{\text{dyn}} = 2$ nPa, and the grey circle represents the position of the Earth within the magnetosphere. Panel b) shows the spacecraft position during the crossing in latitude over longitude distribution with a bin size of 1x1°. Panel c) shows the histogram of the crossing probability, which represents the quality of the MPC as defined in Grimmich et al. (2023a) for all MPCs. The MPCs are colour coded according to their crossing probability: MPCs with a high probability ($> 0.75$) are in green and those with a low probability ($\leq 0.75$) are in blue.

into the aGSE system. The use of this average solar wind velocity for the aberration introduces a small but not drastic bias into the data (e.g., as defined in Grimmich et al., 2023a). This bias may result in a residual aberration effect remaining in the data.

     Most of the crossing locations found are consistent with the shape of the displayed Shue et al. (1998) MP model, except for the crossings found tailward of $x_{\text{agse}} = -10$ $R_E$ (7,817). These are mostly ($\sim$70 %) associated with rather low crossing probabilities ($\leq 0.75$), suggesting a lot of ambiguity in the data that would lead to misidentifications and could explain the 170    apparent deviations. A possible explanation could be that our algorithm confuses the cold dense plasma sheet with solar wind measurements, leading to a change in the assigned data labels and subsequent crossing identification. Thus, we consider the identified crossings tailward of $x_{\text{agse}} = -10$ $R_E$ as less reliable.

     Overall, we found that 20,713 ($\sim$55 %) of the MPCs, of which 12,021 (8,692) are found in the C1 (C3) data, have high crossing probabilities ($> 0.75$). These are used in the following as well-defined crossings. We can also confirm that our database 175    covers not only the equatorial plane, but also a wide range of latitudes and longitudes (see Fig. 2b), in particular with 8,859 well-defined crossings found in the high-latitude regions above $\pm 30°$, as expected from Cluster's orbital coverage. In the appendix, we show two example plots of the Cluster data with the results of the identification process.

In addition, we associate each MPC with time-shifted high-resolution OMNI data at a cadence of 1 min (King and Papitashvili, 2005) to monitor the upstream conditions of the solar wind at the bow shock nose. Various solar wind parameters from the OMNI dataset are taken as the mean values in an 8-minute interval preceding the crossing, if up to 75 % of the data points are available in that interval. The length of the interval chosen takes into account the time delay from the bow shock to the MP and terminator. To estimate this time delay, we assumed typical distances between the bow shock and the MP of 3 to 4 $R_{\mathrm{E}}$, typical distance form the MP to the terminator of 10 $R_{\mathrm{E}}$ and typical flow velocities in the subsolar (flank) magnetosheath of 100 km/s (300 km/s). Using these numbers to calculate how long it would take a plasma element to travel from the BS to the MP and then to the terminator gives the 8 minutes we considered as the time delay. Nevertheless, we assume a stationary and instantaneous response of the MP to the OMNI solar wind conditions.

We use the appropriate dynamic solar wind pressure $p_{\mathrm{dyn}}$ and IMF component $B_{z,\mathrm{IMF}}$ from the corresponding OMNI data to map the observed radial distance $r$ between the Earth and the spacecraft at each well-defined crossing location to an equivalent stand-off distance $R_0$. The mapping is done by converting the functional form of the Shue et al. (1998) model (SH98) to $R_0$, which is then used to compare the observation with the prediction of the SH98 model by calculating a deviation from the theoretical model stand-off distance $\Delta R_0$:

$$R_0 = r \left( \frac{2}{1 + \cos\zeta} \right)^{-\alpha}, \tag{2}$$

$$\Delta R_0 = R_0 - R_{0,\mathrm{SH98}}, \tag{3}$$

where $\zeta$ is the zenith angle between $r$ and the $x$-axis (denoted by $\theta$ in Shue et al., 1997, 1998). The flaring parameter $\alpha$ in (2) and $R_{0,\mathrm{SH98}}$ in (3) are calculated according to equations (11) and (10) in Shue et al. (1998). With sufficient OMNI data available, we can calculate (2) and (3) for 15,781 of the well-defined 20,713 MPCs, giving us a good coverage of the occurrence of crossings under different solar wind conditions.

The mapping of the observed MP location to the stand-off distance $R_0$ is chosen to be comparable to the similar study of the THEMIS dataset in Grimmich et al. (2023a). Furthermore, simply using the radial distance between the modelled MP and the observed MP can introduce a significant bias towards a higher deviation. Due to the curvature of the MP surface, the difference vector between the radial position of the spacecraft and the modelled MP position is not necessarily perpendicular to the surface, resulting in apparently large deviations. This is avoided by comparing the stand-off distances. However, this method can also introduce bias, as any change in MP position is reflected by a global expansion or compression of the entire MP surface, thus ignoring variations caused by tail flaring. Since we will be focusing mainly on the dayside of the magnetosphere, this should not affect our data in any significant way.

Since the cusp region exists as a separate plasma region between the magnetosheath and the magnetosphere and is populated by magnetosheath-like plasma (Lavraud et al., 2004; Pitout and Bogdanova, 2021), our detection algorithm most likely identifies the inner boundaries of the cusp as MPCs. This inner boundary forms a funnel and, when viewed as a traditional MP, causes an indentation in the MP surface (e.g., Lavraud et al., 2004). The SH98 model does not include this cusp indention and therefore may introduce a noticeable bias for MPCs in the cusp region. It would be most noticeable for MPCs observed closer to Earth than predicted (i.e. for negative $\Delta R_0$) due to the detection of the inner cusp boundary. Despite that, the use of this

simple and often used model allows us to make a comparison with the result of Grimmich et al. (2023a) especially in or near the equatorial plane, that is, for crossings with latitudes between $\pm 30°$. In addition, we attempt to quantify the bias introduced by the SH98 model.

Under typical external conditions the cusp should be located between 70° and 85° of magnetic latitude (MLAT) and between 10 and 14 magnetic local time (MLT) (e.g. Pitout and Bogdanova, 2021). Thus, the spacecraft position during the MPC observation is transformed into Solar Magnetic (SM) coordinates, with the z-axis aligned along the Earth's dipole axis (see Laundal and Richmond, 2016, for more details). This allows us to calculate the MLAT and MLT position of each crossing, showing which crossings occur in the area where the cusp is most likely to be located. We define this as the area where

|MLAT|> 70° and |MLAT|≤ 85° and MLT≥ 10 and MLT≤ 14 holds. A total of 593 MPCs (383 well-defined) fully meet the criteria and fall between the MLAT and MLT areas, most likely related to the cusp location. Thus, for only about 2 % of the crossings found, the comparison with the MP model could be affected by a cusp indention bias.

## 3  Database validation and comparison

In order to validate our database, we use preliminary results of the Geospace Region and Magnetospheric Boundary (GRMB)

dataset currently under development (Grison et al., 2024). This dataset aims to have a continuous labelling of the different plasma regions crossed by the Cluster spacecraft during the whole mission duration, using a selection by eye approach.

The GRMB methodology has been validated by ESA in the frame of the contract no. 4000139126/22/ES/CM. It is therefore beyond the scope of this paper to present the full methodology developed to build the dataset. However, we provide some useful information about the magnetopause crossings identification in the dataset. The main criteria to identify to select a

magnetopause region is the change of a highly-variable magnetic field displaying a large wave activity (magnetosheath region) to a slowly varying dipolar magnetic field (magnetosphere region). The identification also relies on changes in the particle population between the magnetosheath and the magnetosphere. Three GRMB items contain magnetopause crossings. IN/MP (sharp MPCs) and IN/MPTR (long, multiple or complex MPCs) should always contain at least one crossing (these items include few minutes of the magnetosheath and magnetosphere on each side). The third item (IN/POL standing for polar regions,

including mainly the cusp signatures) can include crossings when the observed properties switch from magnetosheath to cusp (typical energy dispersion, diamagnetic effect on the magnetic field, particle injections ...). In many occasions IN/POL do not include any crossings: during mid-altitude cusp crossings or where a magnetopause is observed between the magnetosheath and the polar regions. The magnetopause crossing observations have been validated in the frame of the GRMB project by comparing crossings obtained for C3 with reference crossings from the Petrinec et al. (2023) dataset. Limiting the crossings

to October 2003 and March 2007 83 of the 92 reference magnetopause crossings (90 %) are found in one of the three labels IN/MP, IN/MPTR, IN/POL. In half of the mismatches such item was found within 10 minutes. The remaining mismatches were found to correspond to short back and forth crossings that are not included in the GRMB dataset.

We compare the GRMB labels in years 2003 to 2005 and 2007 to 2008 with the outputs of our detection method. For our C1 (C3) dataset, we find that in 77 % (71 %) of the cases where the GRMB indicates an observation of the MP or a transition layer

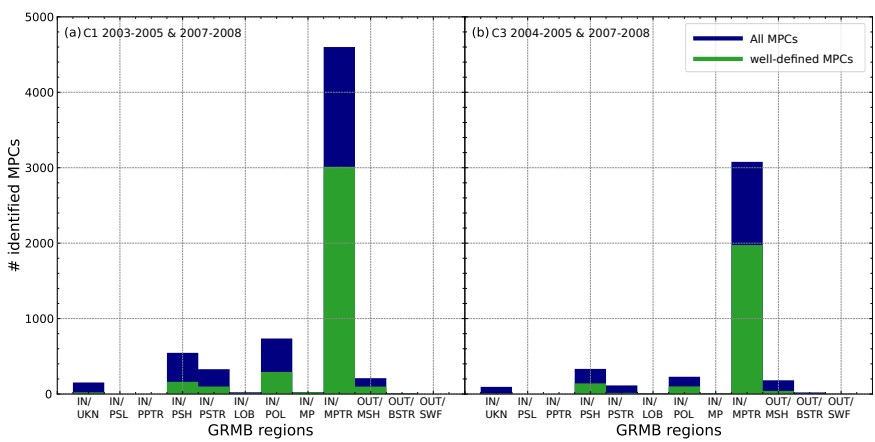

**Figure 3.** Comparison of our identification results with the Geospace Region and Magnetospheric Boundary (GRMB) dataset currently being developed by Grison et al. (2024). The distributions in panel (a) for C1 and in panel (b) for C3 show the number of identified MPCs in the different regions indicated by the GRMB. The labels are an indication of the region where Cluster spacecraft are most likely to be: IN/UKN indicates "Inside the magnetopshere", IN/PLS indicates "Plasmasphere", IN/PPTR indicates "Plasmapause Transition Region", IN/PSH indicates "Plasmasheet", IN/PSTR indicates "Plasmasheet Transition Region", IN/LOB indicates "Lobe", IN/POL indicates "Polar Regions", IN/MP indicates "Magnetopause", IN/MPTR indicates "Magnetopause Transition Region", OUT/MSH indicates "Magnetosheath", OUT/BSTR indicates "Bow Shock Transition Region" and OUT/SWF indicates "Solar Wind and Foreshock". The distribution in orange belongs to all identified MPCs and the distribution in green belongs to the well-defined MPCs with high crossing probabilities (see text and Fig. 2c for details).

with multiple or complex crossings, our identification method also finds at least one crossing. The missing cases are probably due to our pre-selection of appropriate intervals for identifying MPCs and from the continuous GRMB labelling, which also includes the periods when plasma moments or magnetic field data are not available.

We also consider in Fig. 3 the number of crossings (well defined MPCs in green, all MPCs in orange) found in the different regions indicated by GRMB. Here it is also obvious that most of the crossings identified especially the well-defined ones are associated with the IN/MPTR region from the GRMB.

In addition, it is worth to note that the regions with no direct boundary with the MP (IN/PLS indicating plasmasphere, IN/PPTR indicating plasmapause, OUT/SWF indicating solar wind or foreshock) do not contain any crossings. Nevertheless, our identification finds many crossings in magnetospheric regions adjacent to the MP boundary populated with high-energy particles (IN/PSH or plasmasheet and IN/PSTR or plasmasheet transition layer). First, the GRMB dataset does not capture crossings with short back-and-forth changes from one region to another, which could explain why some of our crossings are located in the neighbouring regions of MP (IN/PSTR, IN/PSH, IN/MSH indicating magnetosheath, IN/UKN indicating an ambiguous region inside the magnetosphere). Second, many of these crossings are associated with locations tailward of $x_{\mathrm{agse}} = -10\ R_{\mathrm{E}}$. As noted above, we already consider these crossings to be much less reliable due to the associated low crossing probabilities and the possible relationship with the cold dense plasma sheet (discussion of Fig. 2a), and this validation

seems to confirm this. Since further validation would be needed to use them with confidence, we decided to neglect all MPCs found tailward of $x_{\mathrm{agse}} = -10 \; R_{\mathrm{E}}$ for the moment.

The IN/POL region defined in the GRMB dataset contains magnetosheath-like plasma observed inside the magnetosphere and can be observed close to the MP (distant polar regions). In this, the cusp regions should be included. Therefore, Fig. 3 seems to confirm that only few cusp encounters might be identified as MPC, as in the IN/POL bin we have only a few hundred

crossings identified.

As a summary the GRMB dataset supports our crossing classification. Most of the discrepancies between the two datasets can be explained by the two different approaches (continuous bye-eye selection of the GRMB vs. automatic classification with mandatory input values of the machine learning algorithm).

In a second step we compare all of our well-defined dayside MPCs found near the equatorial plane, that is, crossings where

the latitude position is between $\pm 30°$ and $x_{\mathrm{agse}}$ spacecraft position is positive, with the crossings found in the THEMIS data in the same region by Grimmich et al. (2023a). Figure 4 shows the distributions of these crossings over (a) the MP stand-off distance $R_0$, (b) the deviation from the SH98 model $\Delta R_0$, (c) the spacecraft latitude $\theta$ (d) and longitude $\phi$ position for our two Cluster datasets, the Grimmich et al. (2023b) THEMIS dataset and also for the Petrinec et al. (2023) dataset, which was constructed by visual inspection and is available via the Cluster Science Archive (CSA, Laakso et al., 2010). The distributions

of the different datasets are first normalised with the spacecraft dwell time in the time ranges used for identification to remove observational bias due to spacecraft orbits, and then normalised a second time to important values to better compare these differently sized datasets. The following figures therefore show distributions of the normalised number of crossings per hour.

Overall, the distributions of our Cluster datasets seems to be consistent with the THEMIS and the CSA Cluster datasets. The average stand-off distance $R_0$ of the MP is around 11 $R_{\mathrm{E}}$ for both Cluster sets, matching the 10.5 $R_{\mathrm{E}}$ of the THEMIS

set and the 11 $R_{\mathrm{E}}$ of the CSA set. Nevertheless, we see that Cluster is less likely to encounter $R_0$ values between 8 and 10 $R_{\mathrm{E}}$ in comparison to THEMIS. In general, THEMIS and both Cluster datasets are in agreement with the prediction of the SH98 model within the error bounds. However, crossings further sunward than predicted by the model seems to be slightly more common in the Cluster datasets. The longitude distribution also indicates that Cluster encounters more crossings on the flanks than in the subsolar magnetosphere at the equatorial plane, similar to the THEMIS observations, although the dawn-dusk

asymmetry reported by Grimmich et al. (2023a) is not clearly visible in the Cluster data.

Together with the results from the comparison with the GRMB, the distributions of Fig. 4 give us confidence in our dataset and we consider all well-defined crossing to be valid for the statistical representation of the MP.

In Fig. 5 we show the distributions of our well-defined crossings found within different regions of the magnetosphere over the same parameters as in Fig. 4. If we look at the MPCs that lie in the region where the cusp is expected, we see that $\Delta R_0$,

the deviation of the observed from the SH98 model stand-off distance, is clearly dominated by values around a mean (median) value of -2.05 (-2.04) $R_{\mathrm{E}}$ (Fig. 5b). In these cases, the MP is significantly closer to Earth than predicted, which can be explained by the missing cusp indention in the SH98 model and represents the cusp indention bias mentioned above. Assuming that all these crossings are indeed caused by a cusp encounter, the value of 2 $R_{\mathrm{E}}$ can be considered as an average cusp indentation

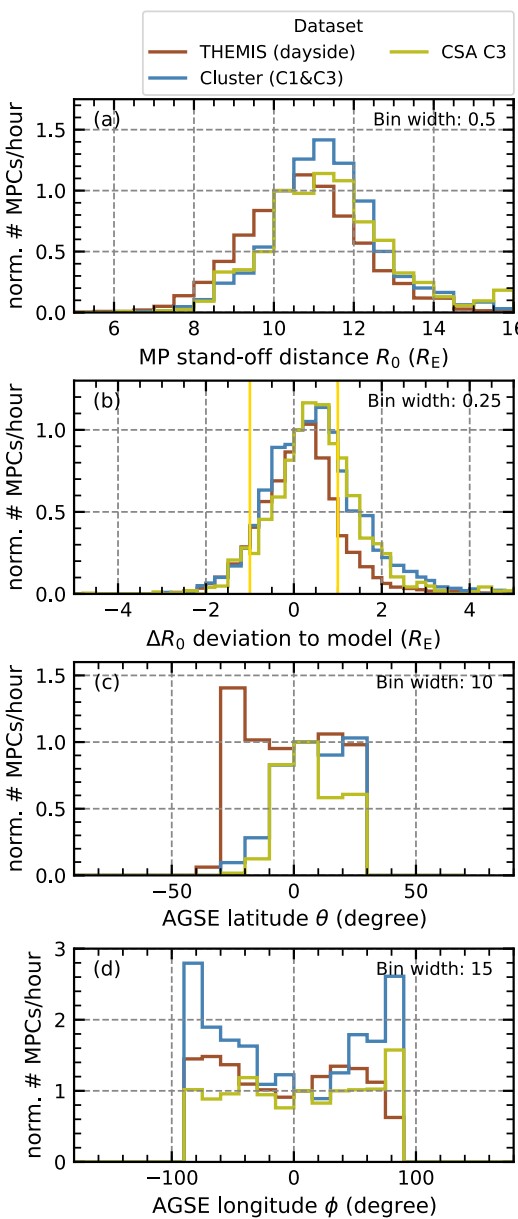

**Figure 4.** Distribution of detected MPCs in the equatorial plane on the dayside for different datasets. The Cluster dataset constructed for this study in blue is compared to the THEMIS dataset from Grimmich et al. (2023a) and another Cluster dataset from Petrinec et al. (2023) in reddish brown and olive respectively. The panels show, from top to bottom, the MP stand-off distance (normalised to the bin 10.0-10.5 $R_E$), the deviation of this distance from the SH98 model distance (normalised to the bin 0.0-0.25 $R_E$), the latitude (normalised to the bin 0°-10°) and longitude (normalised to the bin 0°-15°) at the observation site in aGSE coordinates. The yellow lines in panel (b) represent the uncertainty of the SH98 model.

depth, similar to previously reported depths of 2.5 $R_E$ (Šafránková et al., 2002, 2005), and as a bias value to consider in this region when looking for (extremely) displaced crossings with respect to the SH98 model prediction.

Figure 5 also shows that not only the equatorial MPCs, but the entire dayside crossings are consistent with the SH98 model, with their distribution maximum well within the reported errors of the model (Case and Wild, 2013; Staples et al., 2020) although slightly shifted to negative values around $\Delta R_0 = -0.5 \ R_E$. This shift could be an effect of different solar cycle influences, as the Cluster measurements are mainly from the 23rd cycle, whereas the THEMIS measurements are from the 24th cycle and will be discussed in more detail later. For the crossings in the high latitude regions ($|\theta| > 30°$), on the night side ($|\phi| > 90°$) and also slightly on the flanks ($|\phi| > 30°$), for all latitudes $R_0$ seems to be smaller and overall the MP is found closer to Earth more often than predicted by the SH98 model, although the maximum is still within the model's error bounds. This is partly due to the cusp encounters and the associated indention bias. Nevertheless, it is noteworthy that the agreement between model and observation in these regions is surprisingly good, despite the simplicity and weaknesses of the SH98 model, such as the forced rotational symmetry of the MP surface and the lack of dependence on the dipole tilt for higher latitudes.

Therefore, keeping this cusp bias in mind and focusing only on dayside crossings, we expect the SH98 model to be generally very adequate for further comparisons and identification of MPCs that deviate beyond the errors of $\pm 1 \ R_E$, as has been done by Grimmich et al. (2023a) for the THEMIS data. In the following, crossings occurring outside the cusp region are defined as unusually expanded or compressed MPCs if the deviation from the SH98 model $\Delta R_0$ is greater than 1.5 $R_E$ or less than -1.5 $R_E$, respectively. In total, we find 581 expanded MPCs and 1,739 compressed MPCs on the dayside. Of these, the unusually expanded MPCs are mainly found in the equatorial plane in the subsolar region, while the compressed MPCs are more common at higher latitudes and at the flanks (not shown).

## 4  High latitude magnetopause boundary analysis

We take advantage of the four spacecraft tetrahedron constellation of the Cluster mission and use an automated multi-spacecraft timing method (introduced as eq. 10.20 in Schwartz, 1998) to analyse the MPCs by calculating their normal direction and boundary velocity. We use the 5 vectors per second high resolution magnetic field data from all four Cluster spacecraft at three different 150 s intervals around identified crossing timestamps to find the optimal results in a fully automated way. In each interval, we define a time lag between the crossing observations at the different spacecraft locations by cross-correlating the magnetic field components. The interval and time lag for the timing method is chosen for the interval for which the cross-correlation gives the highest mean correlation coefficients. In the 645 cases ($\sim 3 \ \%$) where we do not have magnetic field data from all four spacecraft, we use a modified timing method using measurements from only three spacecraft combined with coplanarity and related single spacecraft methods (eq. 10.21 in Schwartz, 1998).

The results of the timing method are modified so that the sign of the calculated normal directions points upstream (i.e. with a positive $x$ component) and the sign of the boundary velocity is adjusted accordingly. This modification, together with the assumption that the spacecraft position is fixed, implies that the inbound crossings from the magnetosheath into the magnetosphere should have positive normal velocities, since the MP should be moving in a sunward direction for a fixed spacecraft to

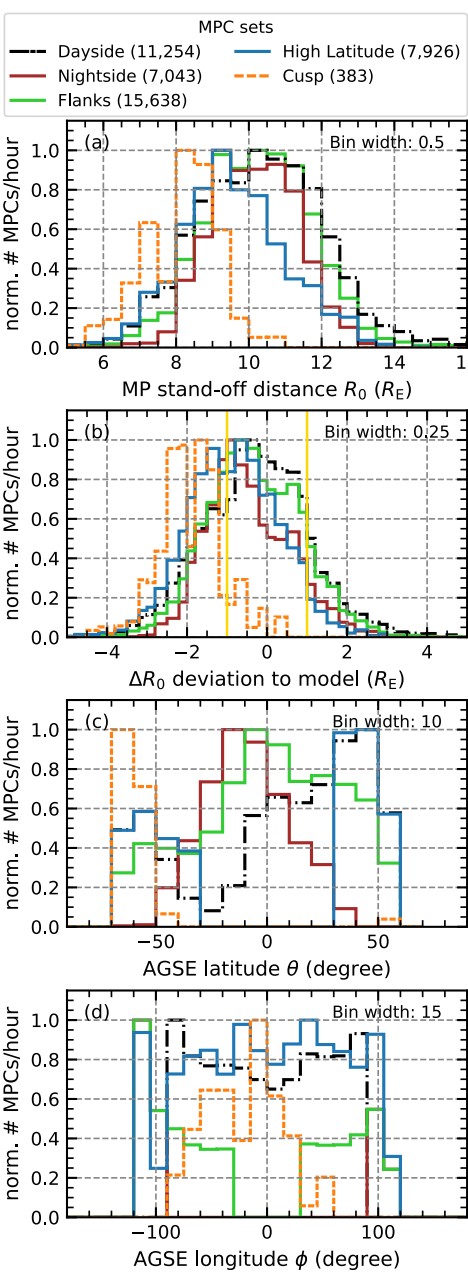

**Figure 5.** Normalised occurrence rates of detected MPCs in the Cluster dataset for different regions of the magnetosphere. The panels are the same as in Fig. 4, with the yellow lines in panel (b) still representing the uncertainty of the SH98 model. Shown are the distributions of the dayside magnetosphere (black), nightside magnetosphere (brown), the flanks (light green), the high latitude regions (blue) and the region where the cusp is most likely to be encountered (orange). To make the different regions more comparable, the normalisation was performed first on the spacecraft dwell time and then on the maximum of each distribution.

cross into the magnetosphere. Subsequently, the outbound crossings from the magnetosphere into the magnetosheath should have negative velocities, since the MP must be moving Earthwards for a fixed spacecraft to cross into the magnetosheath. In line with these definitions, the following discussion of MP velocity will refer to inward MP motion with respect to outbound crossings and outward MP motion with respect to inbound crossings.

Since the geometry of the Cluster spacecraft constellation affects the accuracy of the timing results, we use the planarity $P$ and elongation $E$ of the constellation from Cluster's auxiliary data package to remove events where the timing method could fail. A value of $P = 1$ would indicate that the spacecraft are in a single plane, and for $E = 1$, $P$ is undefined because the spacecraft are in a straight line constellation (see Robert et al., 1998, for further details). In cases where $E$ and $P$ tend towards extremes, the method is highly sensitive to small changes in the time difference between observations, resulting in large errors (Knetter, 2005). Therefore, we use the cut-off threshold of 0.8 for $P$ (if $E \neq 0$) and 0.8 for $E$ in order to avoid larger errors in our results. These constraints leave 6,321 dayside crossings where tetrahedron geometry is preferable to reduce errors in the results.

In addition, for our analysis of the MP normals, we only consider the crossings where the minimum cross-correlation coefficient from the correlation used in the timing method between the different spacecraft measurements is greater than 0.65. This should ensure that the results of the timing method for the remaining 2,117 MPCs are valid as the 4 different measurements are well correlated. Furthermore, we neglect 579 duplicate crossings identified in the C1 and C3 data if they are part of the same observation of the MP, that is, if they have similar timestamps (up to 2 min apart) and yield the same timing method results. MPCs with inconsistencies between the identified crossing type (inbound and outbound) and the calculated MP velocity sign are also neglected, leaving us with only 1,009 unique dayside crossings with well-calculated normals and consistent MP velocities.

In order to get some information as to whether the MP boundary is closed or open and allows magnetic flux to be transported through it, we want to compare the MP normal with the MP adjacent magnetic field in the magnetosheath. We estimate the magnetosheath field by taking the magnetic field vector from the timestamp immediately before and after the MPC timestamp from the time series used for identification process. Based on the label given by our machine learning algorithms, we can identify which of these two vectors belong to the magnetosheath. By calculating the angle between this vector and the MP normal, we can then get a very rough idea of the flow direction of the magnetic flux adjacent to the boundary. Obviously, by taking only a single vector from our resampled time series, we can only give a rough estimate of the magnetic field, which means that our results are only a supported guess about the nature of the boundary.

In the following, we will focus on the 682 MPCs in the high-latitude regions of primary interest, covering 60 % of Cluster's dayside crossings selected using the latitude position threshold of $|\theta| > 30$. Thus, unless otherwise specified, when we refer to MPCs, we mean high latitude crossings (the distribution for all MPCs on the dayside is included in the following histograms for reference purposes only). We want to investigate the MP boundary for different subsets. The first subset gathers the crossings founds where the cusp is most likely to be located (28 MPCs) using the MLAT and MLT thresholds from above. These crossings will be referred to as cusp MPCs. The second subset includes the unusually compressed MPCs, where the observation implies an MP location more than 1.5 $R_{\mathrm{E}}$ Earthwards from the SH98 model prediction (180 MPCs). The third subset includes the

unusually expanded MPCs, where the observation implies an MP location more than 1.5 $R_{\mathrm{E}}$ sunwards from the SH98 model prediction (4 MPCs). In what follows we show the distribution of the first and second subsets with respect to the high latitude crossings and the whole dayside. Given that we could only use 4 of the unusually expanded MPCs to compare with the other subsets, we decided not to show their distributions.

Figure 6 shows the results as a comparison with the SH98 model normals (a) and the magnetic field in the magnetosheath just outside the MP (b), and the overall distribution of MP velocities (c). The distributions shown are individually normalised (see Fig. 6 caption) for better comparison of the subsets. This normalisation is different from the one used in the previous figures. Now only the number of crossings in a particular bin is used for normalisation. The bin used for normalisation is arbitrarily chosen according to expected or theoretically known values, e.g. we expect that the angular deviations between MP normals are only 5° or that typical MP velocities are around 50 km/s.

It can be seen in Fig. 6a that the overall angular deviation of the MP normals from the SH98 model normals tends towards deviations below 35°, with most of the normals showing deviations between 5° and 10°, that is towards agreement between the two normal directions and no undulation of the MP surface. However, larger deviations between 15° and 35° become more dominant for the crossings associated with the unusually compressed MPCs. In these cases, the surface of the MP seems to be more distorted.

The magnetic field vectors in the magnetosheath adjacent to the MP are oriented perpendicular to the MP normals in about 61 % to 66 % of the cases, looking at the bins between 75° and 120 ° (cf. Fig. 6b). In this case, the MP boundary could be associated with a closed boundary where magnetic flux cannot penetrate the MP. Thus, in most cases, the MP motion is a deformation perpendicular to the field lines, probably caused by simple compression or expansion of the magnetosphere. About 12 % of the crossings (with a slightly higher value of up to 16% for the unusually compressed MPCs) show that the angle of the magnetosheath field is more parallel to the MP normals. These crossings would be associated with field line deformation and increase the possibility of magnetic flux penetrating the MP boundary into the magnetosphere, as these crossings are more likely to be associated with an open MP boundary (e.g., Alekseev, 1986; Alexeev and Kalegaev, 1995). Despite expecting the MP boundary to be normally closed, our analysis shows quite a wide distribution for the high latitude MP.

The distribution of MP velocities (Fig. 6c) shows that many of the crossing observations were made at low MP velocities between 0 and 75 $\mathrm{km s^{-1}}$ for both the inward (negative values) and outward MP motion (positive values). It is also clear that the inward MP motion tend to be more often associated with higher velocities with a mean (median) value of -103.4 (-73.6) $\mathrm{km s^{-1}}$ and more often reach very high velocities around -400 $\mathrm{km s^{-1}}$ compared to the outward MP motion with a mean (median) value of 85.0 (65.6) $\mathrm{km s^{-1}}$. And in general we see a tendency to observe more often the inward than the outward MP motion, which is to be expected for the compressed MPCs. Only for the cusp MPCs does the encounter of low velocity outward MP motion seem to be more frequent than inward MP motion. Note that the radial motion along the MP normal may not be the dominant velocity in this cusp encounter, as the cusp moves down in latitude as the solar wind dynamic pressure increases or the IMF $B_z$ component turns southward.

For all well-defined crossings, we also perform a simple Walén test (Paschmann and Sonnerup, 2008) in the same interval used for the optimal timing method, even if the timing result is insufficient. The test determines how accurately the MP can

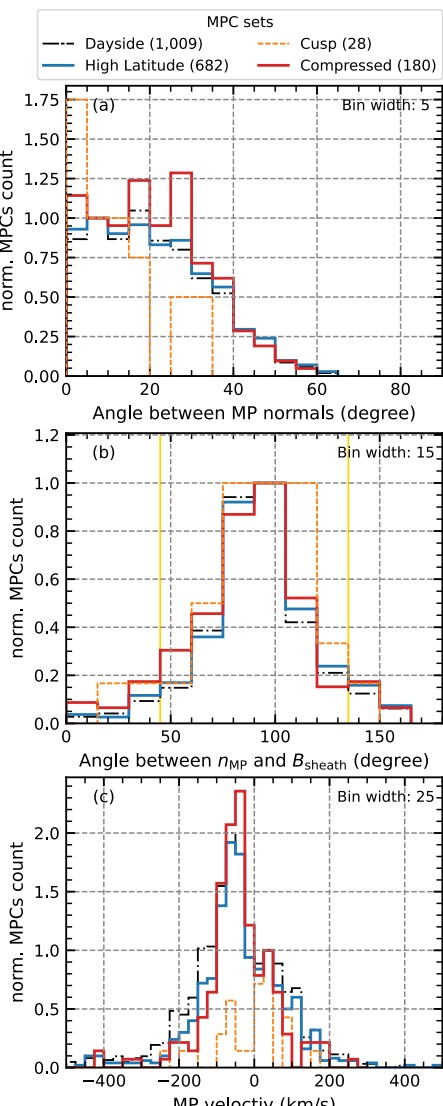

**Figure 6.** Different distributions showing the results derived from the timing method: (a) shows the total angular deviation between the timing estimated MP normal and the predicted MP normal of the SH98 model normalised to the bin between $5°$ and $10°$; (b) shows the angle between the timing estimated MP normals and the magnetic field vectors upstream of the MPC in the magnetosheath normalised to the perpendicular direction in the $90°$ bin. The yellow lines indicate the point from a more parallel to a more perpendicular configuration, corresponding to the point below which the MP is more likely be associated with an open MP boundary; (c) shows the MP velocity distributions normalised to the outbound velocity bin around $37.5 \ \mathrm{kms}^{-1}$. The distribution for all high latitude crossings (blue) is compared with the crossing likely to be associated with the cusp (orange) and with the MPCs for which the observed $R_0$ and SH98 model predictions differ drastically, with only the compressed MPCs shown in red (details in text).

be defined as a rotational discontinuity (RD) based on the fact that the plasma flow immediately upstream and downstream of an ideal RD should be Alfvénic. To implement this test, we use the continuous comparison between the plasma ion velocity transformed into the proper de Hoffmann-Teller (HT) frame

$$\boldsymbol{v}'_{\text{ion}} = \boldsymbol{v}_{\text{ion}} - \boldsymbol{v}_{\text{HT}} \tag{4}$$

and the Alfvén velocity

$$\boldsymbol{v}_{\text{A}} = \sqrt{\frac{1-\alpha}{\rho\mu_0}}\,\boldsymbol{B}, \tag{5}$$

with the factor $\alpha = (p_{||} - p_\perp)\mu_0/\boldsymbol{B}^2$ correcting for the pressure anisotropy between the pressure parallel $p_{||}$ and perpendicular $p_\perp$ to the magnetic field $\boldsymbol{B}$ (cf., Paschmann et al., 2020). Here $\rho$ is the mass density of the plasma and $\mu_0$ is the magnetic constant. The transformation velocity $\boldsymbol{v}_{\text{HT}}$ is calculated using the MP normal $\boldsymbol{n}_{\text{MP}}$ and the velocity $\boldsymbol{v}_{\text{MP}}$ from the timing method, adapting the formula from Liu et al. (2016),

$$\boldsymbol{v}_{\text{HT}} = \frac{\boldsymbol{n}_{\text{MP}} \times ((\boldsymbol{v}_{\text{up}} - \boldsymbol{v}_{\text{MP}}) \times \boldsymbol{B}_{\text{up}})}{\boldsymbol{n}_{\text{MP}} \cdot \boldsymbol{B}_{\text{up}}}, \tag{6}$$

where $\boldsymbol{B}_{\text{up}}$ and $\boldsymbol{v}_{\text{up}}$ are the upstream conditions for the magnetic field and ion velocity respectively. We evaluate the Walén test by fitting a linear regression to the data points of $\boldsymbol{v}'_{\text{ion}}$ versus $\boldsymbol{v}_{\text{A}}$ following

$$\boldsymbol{v}'_{\text{ion}} = w_{\text{sl}} \cdot \boldsymbol{v}_{\text{A}} + \text{offset}, \tag{7}$$

and evaluating the slope $w_{\text{sl}}$ and the associated correlation coefficient $w_{\text{cc}}$ of the fit. The values $w_{\text{sl}} = \pm 1$ and $w_{\text{cc}} = \pm 1$ are considered ideal and indicate an ideal RD under Alfvénic conditions.

The threshold $|w_{\text{sl}}| > 0.5$ is commonly used to identify crossings as RDs. Technically, this threshold could be used as a single quality measure (see discussion in Paschmann et al., 2020). However, we also choose to keep $|w_{\text{cc}}| > 0.7$ to get a higher accuracy on the possible identification. For the 1,009 MPCS on the whole dayside with assumed well-calculated timing results, we find 152 crossings that fulfil the Walén relation and could be the crossings of RDs. In the high-latitude region we find 98 crossings (26 of which are associated with unusually compressed MPCs) where the MP could be considered an RD and the MP motion could be related to reconnection.

From Fig. 7, which shows the results of the timing analysis for the crossings that could be considered RDs according to the Walén test, we can see that the behaviour for the high-latitude crossings is similar to that shown in Fig. 6. The angular deviation from the model normals is dominated by low angles. Most of the time, the crossings are associated with a closed MP boundary, and the MP velocity is distributed between 50 and 100 $\text{kms}^{-1}$ for both the inward and outward MP motion, although more crossings with an inward motion are observed.

However, the few unusually compressed MPCs, which fully satisfy the Walén relation, show a different behaviour, especially in terms of deviation from the model norms (Fig. 7a). Here we can see that large angular deviations around $25°$ dominate, that is for the unusually compressed MPCs events associated with the crossing of an RD the MP is more distorted, compared to the overall high latitude crossings. Also for these crossings, high MP velocity slightly below 200 $\text{kms}^{-1}$ for the outward moving MP are more common than before.

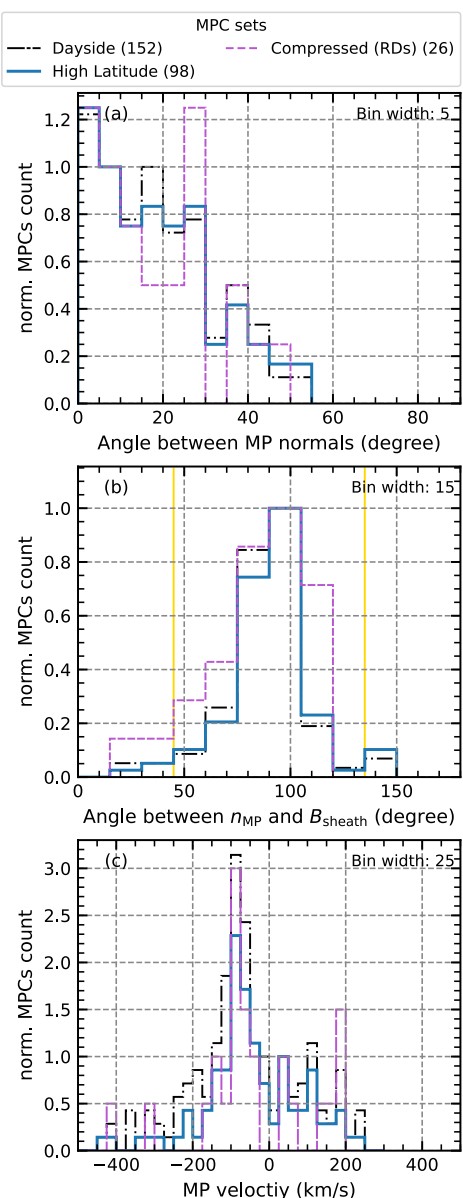

**Figure 7.** Different distributions for events where the Walén relation holds, showing the results derived from the timing method, in the same way as Fig. 6.

## 5 Solar wind influences

The occurrence of MP motion to locations on the dayside that are extremely different from those predicted by SH98 cannot be explained by the dynamic pressure or Bz values of the solar wind. The study of Grimmich et al. (2023a) suggests that in the

**Table 1.** Mann-Whitney U test results: The p-value shown here for all parameters and subsets indicates whether the difference from the general solar wind distribution occurred by chance. A value below 0.05 (marked in bold) indicates that differences do not occur by chance and can therefore be considered statistically significant.

| | Expanded MPCs | Compressed MPCs | Compressed MPCs (RDs) |
|---|---|---|---|
| $B_x$ | $6.5{\cdot}10^{-1}$ | $5.1{\cdot}10^{-3}$ | $1.0{\cdot}10^{-1}$ |
| $B_y$ | $6.5{\cdot}10^{-1}$ | $\mathbf{5.5\cdot10^{-4}}$ | $8.4{\cdot}10^{-1}$ |
| $B_z$ | $3.3{\cdot}10^{-1}$ | $\mathbf{1.0\cdot10^{-2}}$ | $7.0\cdot10^{-2}$ |
| $\lvert B\rvert$ | $2.3{\cdot}10^{-1}$ | $\mathbf{5.1\cdot10^{-3}}$ | $3.6{\cdot}10^{-1}$ |
| $\vartheta_{\mathrm{cone}}$ | $\mathbf{5.6\cdot10^{-13}}$ | $\mathbf{4.7\cdot10^{-12}}$ | $5.1{\cdot}10^{-1}$ |
| $\vartheta_{\mathrm{clock}}$ | $\mathbf{3.6\cdot10^{-2}}$ | $3.9{\cdot}10^{-1}$ | $7.4{\cdot}10^{-1}$ |
| $u_{\mathrm{ion}}$ | $\mathbf{3.4\cdot10^{-9}}$ | $\mathbf{4.1\cdot10^{-7}}$ | $9.2{\cdot}10^{-1}$ |
| $n_{\mathrm{ion}}$ | $\mathbf{6.0\cdot10^{-5}}$ | $\mathbf{3.7\cdot10^{-12}}$ | $\mathbf{3.4\cdot10^{-2}}$ |
| $T_{\mathrm{ion}}$ | $\mathbf{4.3\cdot10^{-9}}$ | $\mathbf{1.5\cdot10^{-4}}$ | $6.5{\cdot}10^{-1}$ |
| $p_{\mathrm{dyn}}$ | $5.9{\cdot}10^{-1}$ | $8.4\cdot10^{-2}$ | $1.3{\cdot}10^{-1}$ |
| $M_A$ | $\mathbf{2.5\cdot10^{-2}}$ | $\mathbf{1.9\cdot10^{-4}}$ | $8.7\cdot10^{-2}$ |
| $\beta$ | $3.2{\cdot}10^{-1}$ | $\mathbf{3.2\cdot10^{-9}}$ | $1.2{\cdot}10^{-1}$ |

equatorial plane, this occurrence is most likely influenced by the IMF magnitude, the IMF cone angle $\vartheta_{\mathrm{cone}}$, the IMF clock angle $\vartheta_{\mathrm{clock}}$, the solar wind bulk velocity $u_{\mathrm{sw}}$, the solar wind Alfvén Mach number $M_A$ and the solar wind plasma $\beta$.

To check whether the high-latitude MP behaves in a similar way, we compare the occurrence distribution of solar wind parameters from the OMNI dataset with the occurrence of solar wind parameters during the observation of the high-latitude MPCs. Once again, we associate the crossings with the average of the OMNI data from an 8-minute preceding interval, as was done before in the calculation for eq. (2). Furthermore, we use all available, well-defined crossings from our dayside dataset (11,252 MPCs), including those that we had previously neglected due to their suspected inadequate results in the multi-spacecraft timing method.

First, a Mann-Whitney U test (Mann and Whitney, 1947) is performed for each variable and subset to determine whether visible differences in distributions are due to chance. Since the solar wind parameter distribution cannot be considered normally distributed, the Mann-Whitney U test is required as a generalisation of the Student's t-test, which assumes the input distribution to be normally distributed. The results of the test on the data are shown in Table 1. Test result values below 0.05 indicate a statistically significant deviation from the general solar wind distribution for the MPC related subset, and therefore the parameters with such values are of primary interest in the following. Conversely, values above 0.05 indicate that the differences occurred by chance, and since this is the case for almost all parameters in the subset containing the compressed MPCs associated with RDs, we do not examine this subset further.

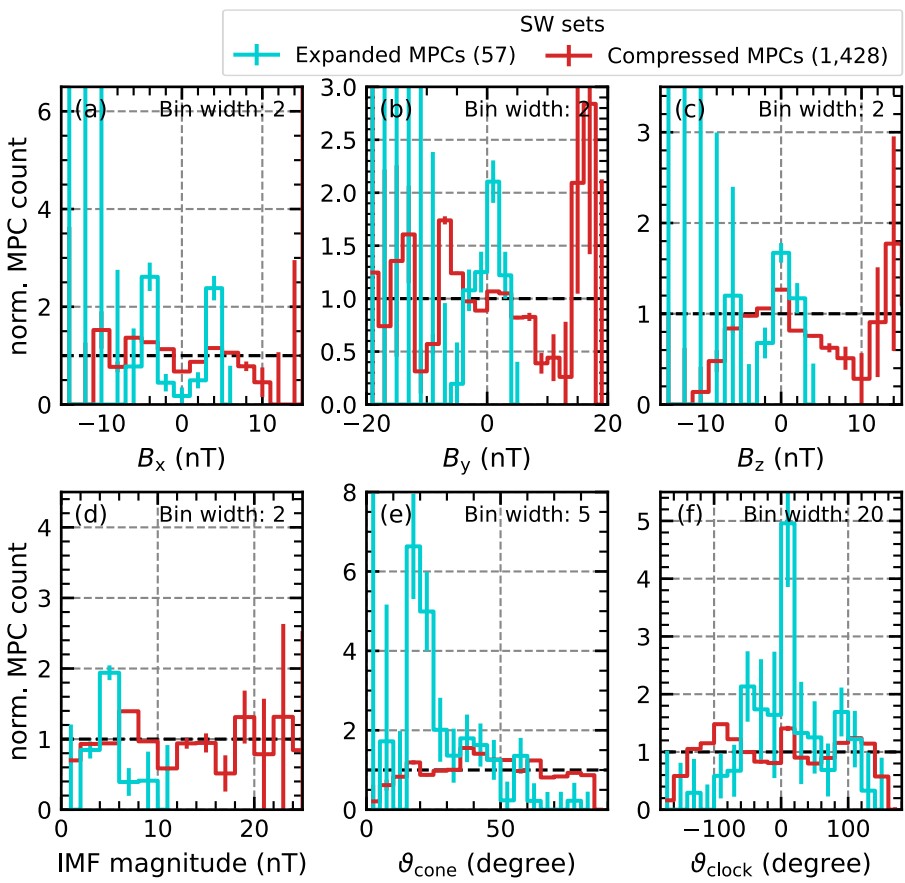

**Figure 8.** Comparison of the distributions of different IMF parameters associated with the observation of MPCs. Each panel shows the distributions associated with the unusually extended MPCs (turquiose), and the unusually compressed MPCs (red). These distributions are normalised by division by the normal solar wind occurrence distribution of the corresponding parameter. Thus, the probability of occurrence would be the same as that of the solar wind at a reference value of one (black dashed lines). For values above one, the occurrence of the different MPCs would be more likely. (a) shows the IMF $B_x$ distributions, (b) shows the IMF $B_y$ distributions, (c) shows the IMF $B_z$ distributions (d) shows the IMF magnitude distributions, (e) shows the IMF cone angle $\vartheta_{\mathrm{cone}}$ distributions and (f) shows the IMF clock angle $\vartheta_{\mathrm{clock}}$ distributions.

Figure 8 and 9 show the comparisons of the distributions with respect to twelve solar wind parameters. To better compare the distributions and see where they deviate from the normal solar wind distributions, we normalized the count rates per bin. The normalisation is done by dividing the parameter distributions associated with the crossings by the natural solar wind distribution in the years 2001 to 2020. Favourable conditions for the occurrence of the observed crossings, especially those deviating from the SH98 prediction, are then visible as quotient maxima above 1 and unfavourable conditions as minima below 1.

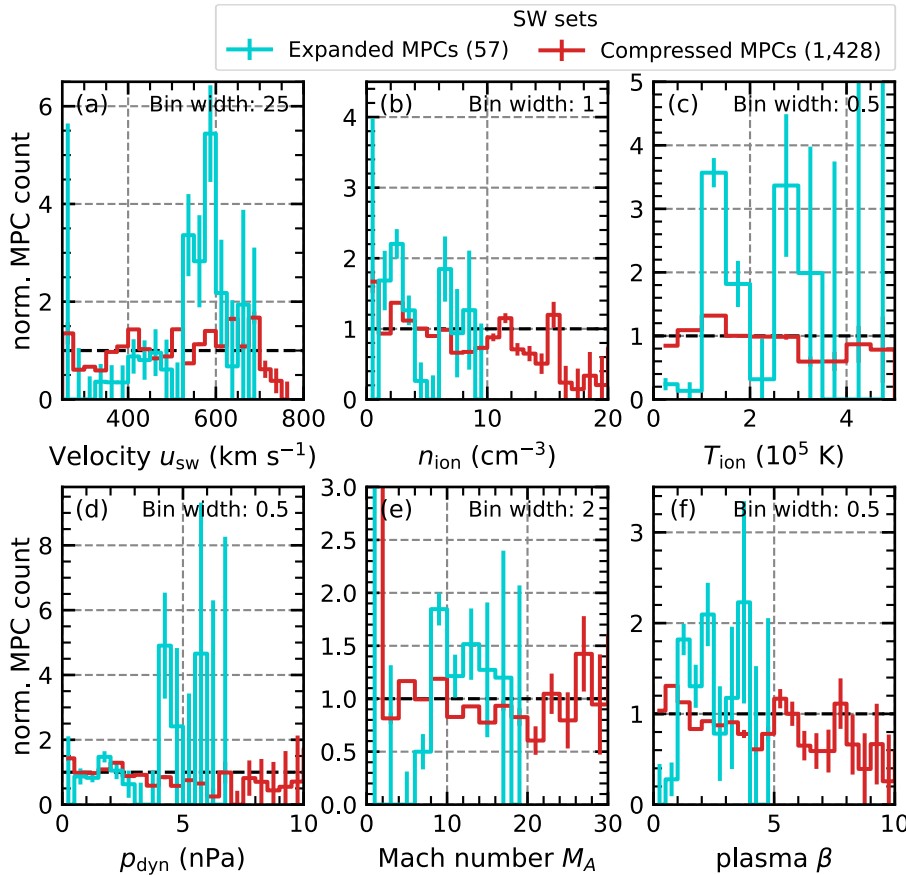

**Figure 9.** Comparison of the distributions of different solar wind plasma parameters associated with the observation of MPCs, similar to Fig. 8. (a) shows the solar wind bulk velocity $u_{\mathrm{sw}}$ distributions, (e) shows the solar wind ion density $n_{\mathrm{ion}}$ distributions, (c) shows the solar wind ion temperature $T_{\mathrm{ion}}$ distributions, (d) shows the solar wind dynamic pressure $p_{\mathrm{dyn}}$ distributions, (e) shows the solar wind Alfvén Mach number $M_A$ distributions and (e) shows the solar wind plasma $\beta$ distributions

From our data set we can extract that on average, two crossings are observed per hour. This is determined by collecting all one-hour intervals in which MPCs are found. We use this average detection rate as a typical identification error and add an estimate of the error to the distributions using

$$\text{bin error} = \frac{2}{\text{data set size} \cdot \text{bin size of solar wind reference}}. \tag{8}$$

This, together with the results from Table 1 on the statistical significance of the deviations, allows us to identify certain parameter ranges where few events are detected and therefore deviations from the reference distribution are not reliable.

In the high latitudes, the distribution of unusually expanded and compressed MPCs show the following behaviour:

Figure 8 shows for the IMF paramters (a) a tendency for the compressed MPCs to favour conditions between 5 nT and 10 nT( panel (d)); (b) a significant influence of $\vartheta_{\mathrm{cone}}$ on expanded MPCs with quasi-radial IMF conditions ($\vartheta_{\mathrm{cone}} < 35°$) clearly

favouring expanded MPCs, while the compressed MPCs are more likely to occur for higher $\vartheta_{\mathrm{cone}}$ around $40°$ (panel (e)); (c) a noticeable deviation in the distribution for the expanded MPCs around low angles, corresponding to occurrences during northward IMF (panel (f)); (d) and frequent occurence of compressed MPCs for $B_y$ values between $\pm 15$ nT and $\pm 20$ nT and $B_z$ values around $0$ nT (panel (b) and (c)).

Figure 9 shows for the remaining solar wind parameters (a) a more frequent occurrence of both expanded and compressed MPCs under high $u_{\mathrm{sw}}$ conditions (above $400$ km $s^{-1}$, cf. panel (a)); (b) more frequent occurrences for both expanded and compressed MPCs when the ion density is around $2$ cm$^{-3}$, and only for the expanded MPCs around $7$ cm$^{-3}$ (panel (b)); (c) that the compressed MPCs are more frequent only for temperatures between $1 \cdot 10^5$ K and $1.5 \cdot 10^5$ K, while the expanded MPCs seem to occur additionally quite often for temperatures around $3 \cdot 10^5$ K (panel (c)); (d) a slight tendency for the occurrence of expanded MPCs under $M_A$ around and above $8$ (panel (e)); (e) and that lower values of the plasma $\beta$ around $1$ seems to lead to more frequent occurrences of compressed MPCs (panel(f)).

Furthermore, we can conclude from Fig. 9d that $p_{\mathrm{dyn}}$ does not show large and significant deviations, which is expected since the effect of this parameter is included in the SH98 MP model. This means that the influences we see from the other parameters are additional influences to the pressure effect and could in some cases weaken this effect, for example at high solar wind speeds.

Studies such as Boardsen et al. (2000); Lin et al. (2010); Liu et al. (2012) have highlighted that the dipole tilt angle $\psi$ can dominate the MP deviations from the SH98 prediction in the high latitude regions. Thus, we also check how $\psi$ influences the occurrence of unusual crossing locations. We calculated $\psi$ as the difference between the orientation of the x/z axis in the SM and the geocentric solar magnetosphere (GSM) coordinates, since $\psi$ describes the orientation of the dipole axis with respect to the Earth-Sun line (e.g., Laundal and Richmond, 2016). According to this definition, $\psi$ is $0°$ when the dipole axis and the Earth-Sun line are perpendicular, and $\psi$ is positive when the dipole pole is tilted towards the Sun.

In Fig. 10 we show the dependence of the MP stand-off distance $R_0$ on $\psi$ and we compare the tilt angles during the observation of the MPCs with the general occurrence of the different tilt angles over the course of the Cluster mission. We can see that in the high latitude region the MP position seems to be influenced by the dipole tilt, as expected. At low tilt angles the observed $R_0$ value is slightly lower than at higher angles (see the linear regression in Fig. 10a). However, the correlation between dipole tilt and stand-off distance is rather weak, due to the large scatter of the crossings. Still, we can see that unusually expanded MPCs are common for $\psi$ around $30°$, while unusually compressed MPCs are more common for angles below $10°$, as suggested by the apparent deviation of the distributions associated with such crossings from the reference (Fig. 10b). The average deviation from the MP model position is about $2.6$ $R_{\mathrm{E}}$ for the low tilt angles and compressed MPCs and $2.2$ $R_{\mathrm{E}}$ for the high tilt angles and expanded MPCs, confirming a significant influence of the tilt angle on the MP position.

## 6 Discussion

The adaptation of the MPC identification method from Grimmich et al. (2023a) was applied to the Cluster data. We use the algorithm from Grimmich et al. (2023a), but trained it on a new dataset to better predict the more complicated plasma regions

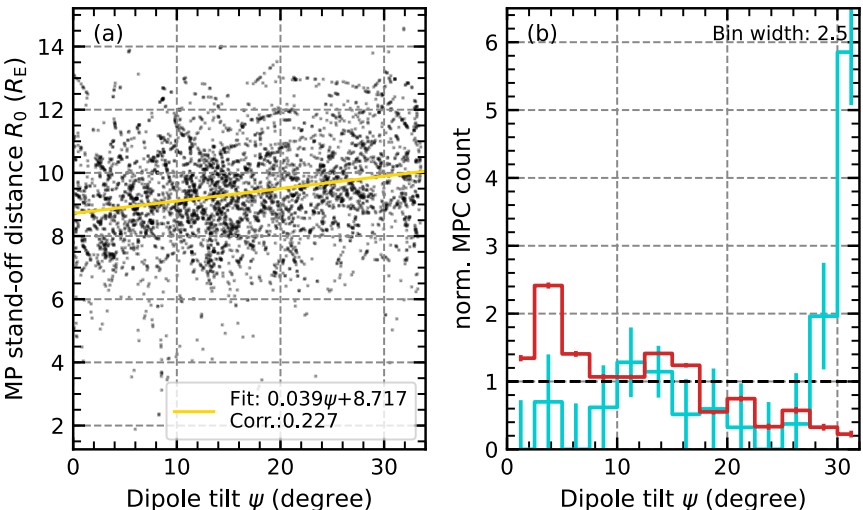

**Figure 10.** Dependence and influence of the dipole tilt angle on the MP position. Panel (a) shows a scatter plot of the observed high latitude MP distance mapped to the stand-off distance $R_0$ versus the tilt angle $\psi$. The yellow line is a linear fit through all the data points, showing a slight dependence with a weak correlation. Panel (b) shows in the same way as Fig. 8 the occurrence of different tilt angles during the observation of an MPC for different subsets (unusually expanded MPCs in turquoise and unusually compressed MPCs in red).

in the high-latitude magnetosphere. Our validation efforts show mostly good agreement of the statistical features associated with the MP by comparing our results with other datasets. However, it is also revealed that the algorithm is probably better suited to finding crossings on the dayside than on the nightside. The reason for this could be that the exclusive use of dayside intervals for the training phase of the algorithm resulted in a tendency to better predict dayside crossings due to over-fitting to

the dayside features. As the nightside crossings can have very different characteristics to the dayside (e.g., Mieth et al., 2019; Raymer, 2018), it is not surprising that the accuracy of our identification on these crossings is reduced. The algorithm may confuse nightside characteristics with dayside magnetosheath features due to over-fitting. Future adaptations of the method should therefore consider this and either use a more diverse training set from both the night- and dayside, or develop separate identification routines for both sides, as was done by Raymer (2018).

With our focus on the high-latitude dayside region of the magnetosphere, the identified crossings show a clear tendency towards low MP stand-off distances compared to the typical value of 10 $R_E$(Fig. 5b), and also lead to the identification of more instances where the MP is closer to Earth than predicted by the MP model (here the SH98 model). We were able to determine that this behaviour is partly due to the encounter with the magnetic cusp, which causes an indentation in the MP surface that is not represented in the SH98 model and therefore shows up in our statistics as lower stand-off distances around

8.5 $R_E$. Using the crossings likely to be associated with the cusp, it is also possible to estimate an average depth of MP surface indentation caused by the cusp. Our estimate of 2 $R_E$ for this depth is in agreement with previous estimates from Šafránková et al. (2002, 2005), with minor deviations due to the fact that we map the crossing observation location to the subsolar point

of an MP SH98 surface fitted to the observed location, whereas Šafránková et al. (2002, 2005) used the direct observation location.

As Cluster's orbit changes over time from a north polar orbit to a south polar orbit, the spacecraft cover both the high latitudes and the equatorial plane. Separating these two regions allowed a direct comparison of the statistics in the equatorial plane with those from Grimmich et al. (2023a), which examines the data from THEMIS spacecraft. In both datasets (Cluster and THEMIS), unusually expanded MPCs are more common in the equatorial plane and around the sub-solar point, while unusually compressed MPCs are found at higher latitudes and on the flanks. Since Cluster operates mainly at high latitude

and on the flanks, and only sparsely at the sub-solar point (cf. Fig. 2b), it is not surprising that we find drastically more of the compressed MPCs in the Cluster dataset. In addition, this difference between the location of occurrence for compressed and expanded MPCs may indicate different processes responding to the occurrence that need to be further investigated in the future.

    It should be noted that the identification of the unusual MPCs is dependent on the MP model chosen, and the limitations of

the SH98 model used here (e.g. the lack of cusp indentation and rotational symmetry) may bias our findings. However, most of the common MP models do not differ drastically in their prediction of the MP position on the dayside (Šafránková et al., 2002; Case and Wild, 2013), so the unusual events should be visible with most of the other models as well.

    The results from the timing method for the general high latitude crossings agree well with previous studies. Plaschke et al. (2009a) examined the deviation of the observed MP normals calculated with minimum variance analysis from the SH98 model

normals on the THEMIS data and finds that most of their events have total angular deviations between $0°$ and $20°$, similar to our findings of primarily small angular deviations from the SH98 model. Larger deviations tend to be more dominant when looking at the compressed MPCs, which is not surprising as such unusually compressed MPCs could be caused by Kelvin-Helmholtz instabilities or magnetosheath jet impacts leading to deformations of the MP surface (Shue et al., 2009; Kavosi and Raeder, 2015; Escoubet et al., 2020; Michael et al., 2021), that is shifting of normal angles with respect to the undisturbed

boundary.

    Furthermore, using the Walén test, we find that 12 % of the high-latitude crossings could be associated with the crossing of an RD and the presence of magnetic reconnection or the encounter of a reconnection-related flux transfer event. These phenomena are associated with inward motion of the MP and deform the MP surface (Aubry et al., 1970; Sibeck et al., 1991; Elphic, 1995; Kim et al., 2024). It is therefore not surprising that the unusually compressed MPCs associated with RDs, which

may result from these phenomena, show larger angular deviations between the estimated and modelled MP normals.

    Our MP velocity distributions with its maxima around $50\,\mathrm{kms^{-1}}$ also agree with the reported most common values of MP motion between $40\,\mathrm{kms^{-1}}$ and $60\,\mathrm{kms^{-1}}$ depending on the investigated regions (Plaschke et al., 2009a; Haaland et al., 2014). Furthermore, in contrast to previous studies our result can distinguish between the inward and outward motion of the MP, showing that in the high latitudes the MP moves outward mostly with velocities between $25\,\mathrm{kms^{-1}}$ and $50\,\mathrm{kms^{-1}}$ and inward

mostly with velocities between $50\,\mathrm{kms^{-1}}$ and $75\,\mathrm{kms^{-1}}$. Although not shown, we also look at the velocity distribution of Cluster MPCs in the mid latitude ranges and find that on average the MP moves inwards at a velocity of $116\,\mathrm{kms^{-1}}$ and outwards at a velocity of $92\,\mathrm{kms^{-1}}$, which is consistent with the finding from Panov et al. (2008), suggesting that the high

latitude MP, with average velocities of 103 kms$^{-1}$ inwards and 85 kms$^{-1}$ outwards, moves more slowly than the mid latitude MP.

As previously reported, the amount of magnetic flux that penetrates the MP boundary when it is open (when the magnetic field in the magnetosheath is more parallel to the MP normal) is of the order of 10 % and may not be a significant contributor to the coupling between the solar wind and the magnetosphere (Alekseev, 1986; Alexeev and Kalegaev, 1995). Our analysis seems to suggest that despite the fact that the MP could be considered closed in many cases, an open boundary is not so rare for high-latitude MPs, especially in about 12 % of the cases where flux penetration is more likely at the MP. Thus, the

penetrating magnetic flux at the dayside high-latitude MP may be more important than expected. However, it is important to note that our calculation and estimation of the angle between the MP normal and the magnetosheath field could be affected by multiple errors due to the automatic calculation of the MP normal direction and a rather simple approach to selecting the magnetosheath magnetic field vector (the measurement from the observing spacecraft 1 min before / after the identified MPC in the magnetosheath).

In general, it is important to note that the all results of the multi-spacecraft timing method should be viewed with caution. We used the automatic selection of the best results by cross-correlation, and cross-correlation showed sufficient correlation between all spacecraft measurements for only 33 % of our MPCs. In many cases, therefore, the timing method will produce very uncertain estimates. Furthermore, although we find a good correlation between the spacecraft measurements and use the constellation geometry parameters to pre-select suitable events, this does not imply good results from the timing method, as

this method can still be strongly influenced by the chosen time difference between the measurements (Knetter, 2005). A few seconds more or less can result in a large angular deviation between the estimated normals of the same event, leading to large errors in normal estimation, especially when using automated detection. However, such automation is necessary for datasets as large as ours.

    The statistical study of the influence of solar wind parameters on the occurrence of unusual crossings in the high-latitudes

shows that the parameters responsible in the equatorial plane (Grimmich et al., 2023a, as reported by) are also important in most cases at high latitudes. This reaffirms the result of Grimmich et al. (2023a): In addition to the influence of the dynamic pressure and the IMF $B_z$ component, quasi-radial IMF conditions with higher Alfvén Mach numbers and ion velocities above 450 kms$^{-1}$ are favourable for magnetospheric expansions beyond the SH98 model predictions, while magnetospheric compressions are associated with more southward IMF conditions with plasma $\beta < 1$ lower ion density and faster solar wind velocities. While

the Mach number effect seems to be less pronounced in the high latitudes compared to the THEMIS observations reported in Grimmich et al. (2023a), the influence of the clock angle and the cone angle seems to be more significant and clearly shows that compressed MPCs occur more frequently during southward and non-radial IMF. This strongly suggests the importance of reconnection-related phenomena at high latitudes in studying the inward motion of the MP.

    We also showed that the tilt angle of the dipole has a significant influence on the MP position, in agreement with the results

of Boardsen et al. (2000) and Liu et al. (2012). However, multivariate analysis is required to determine which of the various solar wind parameters and tilt angle influences are the dominant drivers of the unusual MP displacements.

Note that the Cluster observations come mainly from the period between 2001 and 2009, which corresponds to the declining phase of solar cycle 23 Raymer (2018) found that large compressions of the MP are observed during this declining phase, while the MP is highly inflated during the deep and extended solar minimum and during solar cycle 24 between 2007 and 2014. This minimum is the time when THEMIS observes many of its crossings of the MP. Therefore, there may be a bias towards compressed MPCs in the Cluster data because of the solar cycle phase.

Additionally, it has been previously reported that the distribution of solar wind parameters, such as IMF magnitude and dynamic pressure, varies throughout a solar cycle and across multiple cycles. In contrast, distributions for parameters like the cone angle remain more constant (e.g., Samsonov et al., 2019; Vuorinen et al., 2023). It is therefore not surprising that there are some differences in the conditions that favour the occurrence of unusual MPCs between THEMIS and Cluster, since the observations were made during different solar cycles.

Another bias that may be important to consider here is highlighted in the study by Vuorinen et al. (2023), which reports an uneven coverage of annual solar wind conditions due to variations in spacecraft apogees. This affects the annual occurrence rate of magnetosheath jets, but is also important for other localised observations such as our MPCs. Therefore, the solar wind conditions used for the comparison may not be representative enough, which could also explain the difference between THEMIS and Cluster observations of unusual MPCs. This possibility should be further considered when comparing the influences of different solar wind conditions on MP motion over the years.

It is also important to bear in mind that due to the nature and spatial structure of the solar wind, the conditions measured at L1 (the input of OMNI) may not affect the Earth (Borovsky, 2018; Burkholder et al., 2020). Studies such as Burkholder et al. (2020) or O'Brien et al. (2023) suggest that OMNI's propagation approach (Weimer et al., 2003; King and Papitashvili, 2005) is rather limited and that other approaches may be more useful to better reflect the reported spatial structure. Thus our use of OMNI as input here, and also in the study of Grimmich et al. (2023a), should be seen as an educated guess for the possible influence of different solar wind parameters on the MP motion to unusual locations. In the future, more attention needs to be paid to the input parameters.

Finally, our dataset used in this research was in some cases very limited due to our applied selection and filter criteria, with only a few events available for statistical analysis, especially when looking at the unusually crossing events. The number of expanded MPCs is rather limited compared to the number of compressed MPCs. With the completed GRMB dataset (Grison et al., 2024) an even larger dataset of Cluster MPCs will be available and could be used to verify our findings. In addition, the plasma measurements from CIS-CODIF available on C4 during the study period could be used to identify additional crossings, especially in the times when HIA starts to fail (after 2014).

## 7 Conclusions

In this study, we have presented a new dataset of Cluster magnetopause crossings, collected between the years 2001 and 2020, by adapting the methodology of Grimmich et al. (2023a). Our dataset showed good agreement with other datasets and allowed a detailed study of the high-latitude magnetospheric region.

We found that (1) the high-latitude MP motion is on average faster inward than outward, remaining in general agreement with previously reported values; (2) the boundary appears to be often closed, with about 12 % of cases showing a configuration where the MP could be open, allowing flux penetration across the boundary in these cases; (3) on the dayside, similar solar wind parameters are responsible for the occurrence of MP positions beyond the SH98 model prediction in high latitudes and in the equatorial plane; (4) the dipole tilt angle influence on the MP location is significant in high latitudes and can lead to deviations from the modelled MP position of more than $2\,R_{\mathrm{E}}$.

Since the previous study by Grimmich et al. (2023a) investigated large deviations from the modelled MP position in the equatorial region, and this study now extends this investigation to the high latitudes, we can begin to formulate a more global behaviour of the MP response to solar wind influences beyond the dynamic pressure. However, in addition to the identified external sources and the influence of the dipole tilt angle on the motion of the MPs, it is also important to look at other internal parameters, such as geomagnetic activity, to determine whether they are important. Once all possible sources have been collected, it remains to be determined by a multivariate analysis which parameters and parameter combinations are the dominant source for the occurrence of MP positions beyond the SH98 and other models.

The upcoming SMILE mission (Branduardi-Raymont et al., 2018) will directly infer the shape and location of the MP at multiple latitudes, mostly coupled with in-situ measurements in the magnetosheath and occasionally with measurements in the solar wind, and will encounter the high-latitude MP during each of its orbits. Our study could therefore provide information on how to interpret the SMILE measurements and could also be used to improve the existing MP models needed for SMILE analysis (Wang and Sun, 2022). In addition, the new data from the mission will allow a direct comparison with the results of our study and allow further studies of unusual events.

*Code and data availability.* Cluster data are publicly available via the Cluster Science Archive at https://csa.esac.esa.int/csa-web/ and OMNI data can be accessed via the GSFC/SPDF OMNIWeb interface at https://omniweb.gsfc.nasa.gov. The Open Science Framework (OSF) hosts the assembled MPC database by (Grimmich et al., 2024a) for C1 and C3 at https://osf.io/pxctg/. Access to the GRMB datasets used can be granted by contacting the GRMB team at IAP and BIRA. For long-term preservation, the definitive GRMB dataset and a link to the reference publication will soon be available on the Cluster Science Archive website. Until then, the preliminary GRMB data used in this study can also be found at https://osf.io/pxctg/. The magnetopause crossings by Petrinec et al. used to validate the GRMB and select the training intervals for our study can be found at https://www.cosmos.esa.int/web/csa/bow-shock-magnetopause-crossings. To collect and analyse the spacecraft data, we used the open source Python Space Physics Environment Data Analysis Software (pySPEDAS) by Grimes et al. (2022), which can be found at https://github.com/spedas/pyspedas.

**Appendix A: Examples of magnetopause identification**

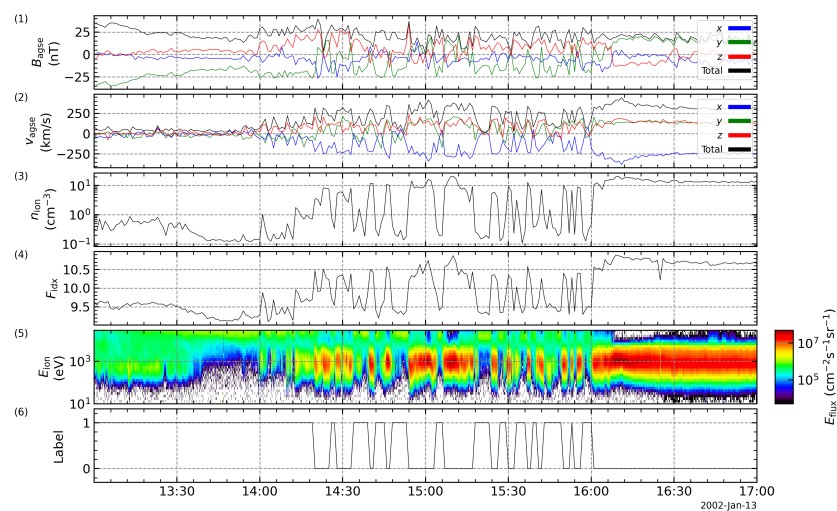

**Figure A1.** Time series of cluster C3 data on 13 January 2002. The spacecraft moves outbound from the magnetosphere into the magnetosheath. From top to bottom the panels show the magnetic field data, the ion velocity, the ion density, the flux index, the energy flux density and the data label given by the machine learning procedure. Changes in the label can be used to identify magnetopause crossings. In this case, the algorithm detects label changes that are quite clear.

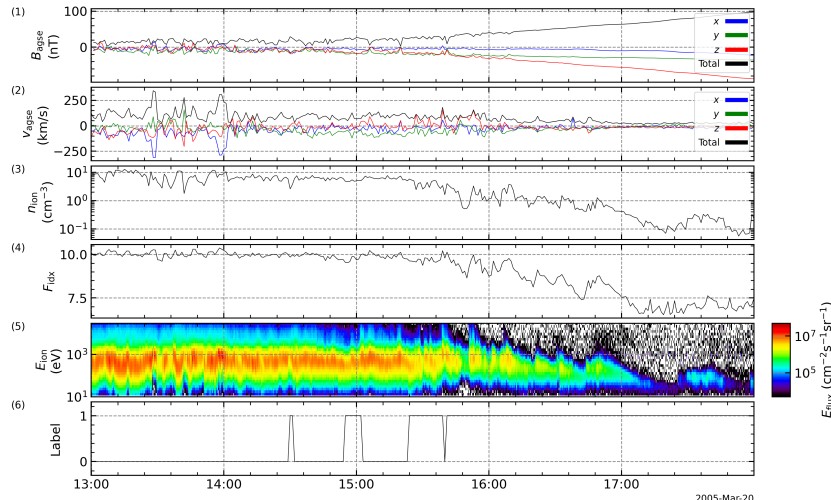

**Figure B1.** Time series of cluster C1 data on 20 March 2005. The spacecraft moves inbound from the magnetosheath into the magnetosphere. From top to bottom the panels show the magnetic field data, the ion velocity, the ion density, the flux index, the energy flux density and the data label given by the machine learning procedure. Changes in the label can be used to identify magnetopause crossings. In this case, the algorithm identifies ambiguous label changes.

*Author contributions.* NG constructed the database, performed the analysis and wrote the original manuscript. FP was involved in developing the research idea for this study and helped to improve the manuscript. BG and FD kindly shared their preliminary GRMB dataset for this study and helped to compare the datasets. FP helped on discussing the potential errors arising from the multi-spacecraft timing method. CPE, MOA, ODC, SH, RN and DGS all helped with the discussion and finalisation of the manuscript. MH, RM are included as part of the development team of the GRMB.

*Competing interests.* The authors declare that the research was conducted in the absence of any commercial or financial relationships that could be construed as a potential conflict of interest.

*Acknowledgements.* This work, particularly NG and FP, was supported by the German Center for Aviation and Space (DLR) under contract 50 OC 2401. The work done by BG, FD, MH and RM was supported by the project GRMB (Geospace Region and Magnetospheric Boundary) under the ESA Contract No. 4000139126/22/ES/CM. MOA was supported by UKRI (STFC/EPSRC) Stephen Hawking Fellowship EP/T01735X/1. This research was supported by the International Space Science Institute (ISSI) in Bern, through ISSI International Team project #546 "Magnetohydrodynamic Surface Waves at Earth's Magnetosphere (and Beyond)". We thank Laakso et al. (2010) for their efforts in providing Cluster mission data and user guides through the Cluster Science Archive (CSA), and acknowledge the work of the

FGM and CIS instrument teams in providing valuable science data. We thank J. King and N. Papitashvili of the National Space Science Data Center (NSSDC) in the NASA/GSFC for the use of the OMNI 2 database. The authors also want to thank Eric Grimes, Jim Lewis and Nick Hatzigeorgiu for the ongoing development of the open source Python Space Physics Environment Data Analysis Software (pySPEDAS) used here to download and process the necessary data.

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
