# Peer review of "The Cluster spacecraft's view of the motion of the high-latitude magnetopause"

_EGUsphere, 2024_

## Author Comment (AC1)

| | Expanded MPCs | Compressed MPCs | Compressed MPCs (RDs) |
|---|---|---|---|
| $B_x$ | $6.5 \cdot 10^{-1}$ | $5.1 \cdot 10^{-3}$ | $1.0 \cdot 10^{-1}$ |
| $B_y$ | $6.5 \cdot 10^{-1}$ | $5.5 \cdot 10^{-4}$ | $8.4 \cdot 10^{-1}$ |
| $B_z$ | $3.3 \cdot 10^{-1}$ | $1.0 \cdot 10^{-2}$ | $7.0 \cdot 10^{-2}$ |
| $|B|$ | $2.3 \cdot 10^{-1}$ | $5.1 \cdot 10^{-3}$ | $3.6 \cdot 10^{-1}$ |
| $\vartheta_{cone}$ | $5.6 \cdot 10^{-13}$ | $4.7 \cdot 10^{-12}$ | $5.1 \cdot 10^{-1}$ |
| $\vartheta_{clock}$ | $3.6 \cdot 10^{-2}$ | $3.9 \cdot 10^{-1}$ | $7.4 \cdot 10^{-1}$ |
| $u_{ion}$ | $3.4 \cdot 10^{-9}$ | $4.1 \cdot 10^{-7}$ | $9.2 \cdot 10^{-1}$ |
| $n_{ion}$ | $6.0 \cdot 10^{-5}$ | $3.7 \cdot 10^{-12}$ | $3.4 \cdot 10^{-2}$ |
| $T_{ion}$ | $4.3 \cdot 10^{-9}$ | $1.5 \cdot 10^{-4}$ | $6.5 \cdot 10^{-1}$ |
| $p_{dyn}$ | $5.9 \cdot 10^{-1}$ | $8.4 \cdot 10^{-2}$ | $1.3 \cdot 10^{-1}$ |
| $M_A$ | $2.5 \cdot 10^{-2}$ | $1.9 \cdot 10^{-4}$ | $8.7 \cdot 10^{-2}$ |
| $\beta$ | $3.2 \cdot 10^{-1}$ | $3.2 \cdot 10^{-9}$ | $1.2 \cdot 10^{-1}$ |

[Figure]

[Figure]

Abstract:

The Cluster mission consists of 4 identical spacecraft, each carrying 11 scientific experiments. The spacecraft were launched in July and August 2000 into near polar inclined, 19x4 RE elliptic orbits. All four spacecraft are still in operation 23 years later. The magnetosphere environment is highly dynamic and its regions cannot be accessed by the orbital information alone. The purpose of this study is to develop a comprehensive dataset, providing information on Geospace Region and Magnetospheric Boundaries (GRMB) crossed by each of the four Cluster spacecraft, and to deliver it to the Cluster Science Archive (CSA).

The GRMB dataset provides a classification useful for the scientific community. Therefore, the methodology does not define what is a bow shock or what is a magnetopause. The goal is to have labeled regions that contain the bow-shocks or magnetopauses. And then each user can apply its own definition on the appropriate label subset. The GRMB list contains two kinds of items:

- **Regions**: Magnetosphere, Magnetosheath, Lobe, Solar wind / Foreshock, Plasmasheet, Plasmasphere

- **Transition regions**: Bow shock TR, Magnetopause TR, Polar regions, Plasmasheet TR, Plasmapause TR

Transition regions can include properties matching several regions. For example, a bow shock TR can include short periods of solar wind or magnetosheath. Solar wind and magnetosheath should not include bow shock crossings.

The GRMB dataset is based on more than 40 data products available at CSA, taken from 7 instrument suites. The methodology relies on the visual identification of the boundaries between two consecutive GRMB items.

The methodology, the criteria applied for the boundary identification, and the dataset validation are presented. The dataset is not yet fully completed but the Cluster location is already available for more than 5 years per spacecraft.

The visualization of the regions, and their physical properties, crossed by the Cluster spacecraft during several years, illustrates the scientific interest of this dataset.

**Analysis of Cluster data with the publicly available GRMB (Geospace Region and Magnetospheric Boundary) dataset**

EGU24-13267

Benjamin **Grison**[1], Fabien Darrouzet[2], Romain Maggiolo[2], Mykhaylo Hayosh[1], Matthew G. Taylor[3]

(1) Institute of Atmospheric Physics (IAP) of the Czech Academy of Sciences, (2) Royal Belgian Institute for Space Aeronomy (BIRA-IASB), (3) ESA/ESTEC

**GRMB list: 15 items**

| Index | Label | Name |
|---|---|---|
| 1 | IN/UKN | INside the magnetosphere |
| 2 | IN/PLS | PLasmaSphere |
| 3 | IN/PPTR | PlasmaPause TR |
| 4 | IN/PSH | PlasmaSHeet |
| 5 | IN/PSTR | PlasmaSHeet TR |
| 6 | IN/LOB | LOBe |
| 7 | IN/POL | POLar Regions |
| 8 | IN/MP | MagnetoPause |
| 9 | IN/MPTR | MagnetoPause TR |
| 10 | OUT/MSH | MagnetoSHeath |
| 11 | OUT/BSTR | Bow Shock TR |
| 12 | OUT/SWF | Solar Wind and Foreshock |
| 13 | OUT/UKN | OUTside the magnetosphere |
| 14 | UNKOWN | Unknown |
| 15 | N/A | Void |

PLS: The plasma frequency is out of WHISPER range close to perigee
PPTR: the plasma density gradient close to perigee
PSH: high energy and isotropic plasma is observed
PSHTR is less homogeneous than PSH
LOB: low-plasma density region
POL: magnetosheath-like plasma (+ energy dispersion) observed inside the magnetosphere
MP: sharp magnetopause crossing
MPTR: complex magnetopause crossings
BSTR: (anti-)correlated abrupt changes in n, B (v) separating MSH and SWF

**Methodology**

[Figure]

Snapshot from the GRMB region selection tool software

**GRMB preliminary dataset: Year 2007 output for Cluster-1**

[Figure]

[Figure]

In 2007, C1 spent most of the time in the LOB, SWF, PSH and MSH regions

[Figure]

*Inbound_vs_outbound* crossings (left). Separation of mid-altitude (in blue) and high-altitude (in grey) polar regions (right).

**Item properties: B and n**

[Figure]

Two-dimensional histograms (2007/C1) of the magnetic field (FGM) divided by the IMF magnitude (OMNI) as a function of the density (CIS/HIA) divided by the solar wind density. The red lines correspond to separatrix between the magnetosphere (left), the solar wind (bottom left) and the magnetosheath (right) as defined by *Nguyen et al.* (2022, JGR).

**Methodology Validation**

The comparisons with reference lists (magnetopause: *Trattner et al.* (ESA) with MPTR/MP/POL; bow shocks: *Kruparova* et al. (2007, JGR) with BSTR; cusps: *Pitout et al.* (2006) with POL; plasmapauses: *Darrouzet et al.* (2013, AnnGeo) with PPTR) show a very good agreement with the GRMB dataset (90%). The main limitation is the minimum resolution of the pre-generated plots (1-hour): some short back and forth boundary crossings are not detected.

The comparison with the automatic list ECLAT (*Boakes et al.*, 2006, JGR) is difficult due to the different time resolution (regions shorter than about 20 min are not resolved in the GRMB dataset).

**IN/MSH insights**

IN/MSH entries are split into 10-minutes interval.

[Figure]

The bow shock normal angle is computed at the C1 location in IN/MSH (see *Soucek et al.*, 2014, JGR)

[Figure]

The BS angle: QuasiPar (0-30 deg), QuasiPerp (60-90 deg) and Intermediate (30-60 deg).

Magnetic power spectral density (PSD) of the STAFF Prime Parameters (PP).

[Figure]

The high PSD in the quasi-parallel sector is probably related to turbulent wave activity.

[Figure]

Comparison of the anisotropy of the wave activity in the 3 frequency bands of the Prime Parameters.

**Magnetic wave activity in IN/MPTR, OUT/MSH , OUT/BSTR , and OUT/SWF**

[Figure]

Entries are split in 10-minutes interval. In each frequency band of the STAFF PP the median value is the highest in the OUT/BSTR, and the lowest in OUT/SWF.

**Label Distribution per days**

[Figure]

**Multi-SC and multi-years comparisons**

**Output: CEF file description**

The CEF file of the GRMB dataset encloses seven variables :

- *time tags* : time interval [start ; end]
- *location_label* : GRMB item short name (string)
- *location_code*: GRMB item index (int)
- *quality_location_label* : displayed panels at selection (string)
- *quality_location_code*: based on product availability (int)
- *inbound_vs_oubound*: inbound or outbound crossing (int)
- *crossing_complexity*: operator's appreciation (bool)

**Summary**

The GRMB dataset will provide a continuous coverage of the Cluster location of each of the four spacecraft for the whole mission duration.

15 labels are considered.

**The first years of the dataset (2001-2007) will be made available via the Cluster Science Archive (https://csa.esac.esa.int/csa-web) in May.**

This dataset aims to support the scientific community not only by providing the location, but also to perform statistical studies. For example, the user can select the BSTR region to apply his/her own selection criteria for bow shock identification inside this transition region.

The realization of the GRMB dataset is funded by ESA Contract No. 4000139126/22/ES/CM Geospace Region and Magnetospheric Boundary Identification using the Cluster Science Archive C1/STAFF/ PP have been downloaded from AMDA (https://amda.irap.omp.eu/)

**Cluster locations in the geospace revealed by the GRMB (Geospace Region and Magnetospheric Boundary) dataset**

Benjamin **Grison**[1], Fabien Darrouzet[2], Romain Maggiolo[2], Mykhaylo Hayosh[1], Matthew G. Taylor[3]

(1) Institute of Atmospheric Physics (IAP) of the Czech Academy of Sciences, (2) Royal Belgian Institute for Space Aeronomy (BIRA-IASB), (3) ESA/ESTEC

Abstract:

The Cluster Mission consists of four identical spacecraft, each carrying 11 scientific experiments. The spacecraft were launched in July and August 2000 into near polar inclined, 19x4 $R_E$ elliptic orbits and all four spacecraft are still in operation. The magnetosphere environment is highly dynamic and its regions cannot be accessed by the orbital information alone. The purpose of study is to develop a comprehensive dataset, providing information on Geospace Region and Magnetospheric Boundaries (GRMB) made for each of the four Cluster spacecraft, and deliver it to the Cluster Science Archive (CSA).

The GRMB dataset aims at providing a classification useful for the scientific community. For example, the methodology does not define what is a bow shock or what is a magnetopause. The goal is to have labeled regions that contain the bow-shocks or magnetopauses. And then each user can apply its own definition on the appropriate label subset.

The GRMB list contains two kind of items:

- **Region**: Magnetosheath, Lobe, Solar wind and foreshock, Plasmasheet, Plasmasphere

- **Transition regio**n (TR): Bow shock TR, Magnetopause TR, Polar Regions, Plasmasheet TR, Plasmapause TR

Transition regions can include properties matching several regions. For example, Bow shock TR can include short periods of solar wind or magnetosheath. Solar wind and magnetosheath should not include bow shock crossings.

The GRMB dataset is based on more than 40 data product available at CSA, taken from seven instrument suites. The methodology relies on the visual identification of the boundaries between two consecutive GRMB items.

The methodology and the criteria applied for the boundary identification are presented. The validation is made by comparing regions crossed by different spacecraft during the same period and by comparing the regions with published list of plasmapause, magnetopause, bow-shock crossings, etc... Comparisons made with output from ECLAT automatic classification method highlights the difference in visual classification and automatic classification.

The visualization of the regions crossed by the Cluster spacecraft during several years illustrates the scientific interest of this dataset. As a first application, the dataset is used to compare physical properties between regions.

**GRMB list: 15 items**

| Index | ShortName | Name |
|---|---|---|
| 1 | IN/UKN | INside the magnetosphere |
| 2 | IN/PLS | PLasmaSphere |
| 3 | IN/PPTR | PlasmaPause TR |
| 4 | IN/PSH | PlasmaSHeet |
| 5 | IN/PSTR | PlasmaSHeet TR |
| 6 | IN/LOB | LOBe |
| 7 | IN/MP | MagnetoPause |
| 8 | IN/MPTR | MagnetoPause TR |
| 9 | IN/POL | POLar Regions |
| 10 | OUT/MSH | MagnetoSHeath |
| 11 | OUT/BSTR | Bow Shock TR |
| 12 | OUT/SWF | Solar Wind and Foreshock |
| 13 | OUT/UKN | OUTside the magnetosphere |
| 14 | UNKOWN | Unknown |
| 15 | N/A | Void |

Basic item description:

PLS: The plasma frequency is out of WHISPER range close to perigee

PPTR: the plasma density gradient close to perigee

PSH: high energy and isotropic plasma is observed

PSHTR is less homogeneous than PSH

LOB: low-plasma density region

MP: sharp magnetopause crossing

MPTR: complex magnetopause crossings

POL: magnetosheath-like plasma (+ energy dispersion) observed inside the magnetosphere

BSTR: (anti-)correlated abrupt changes in n, B (v) separating MSH and SWF

**Methodology**

[Figure]

This is a join project between IPA and BIRA.

Input: CSA pre-generated plots.

Output: Continuous coverage of the spacecraft location.

The plot time resolution is 1-day, 6 hours and 1 hour.

The dataset identifies sharp boundaries.

Otherwise, the time resolution is about 20 min.

Internal check relies on orbit comparisons and the order of region crossings.

**GRMB preliminary dataset: Year 2007 output for Cluster-1**

[Figure]

**Item Selection**

[Figure]

Snapshot from the GRMB region selection tool software

**GRMB items occurrence (C1 Y2007) (All or Merging consecutive identicals)**

**GRMB items occurrence**

In 2017, C1 spent most of the time in the LOB, SWF, PSH and MSH regions

**Output: CEF file description**

The preliminary CEF file of the GRMB dataset encloses seven variables :

- **time tags** :  time interval [start ; end]
- **location_label** :  GRMB item short name (string)
- **location_code**:  GRMB item index (int)
- **quality_location_label** :  displayed panels at selection (string)
- **quality_location_code**:  GRMB item short name (string)
- **inbound_value**:  inbound or outbound crossing (int)
- **crossingcomplexity_value**:  operator's appreciation (bool)

[Figure]

*inbound_value* is defined in accordance with the Cluster orbit

**IN/POL Inbounds and Outbounds**
**C1 2007 ( Orbits: 1001 - 1154 )**

- n/a
- inbound
- outbound
- back and forth

[Figure]

*inbound_value* also allows to separate mid-altitude (in blue) and high-altitude (in grey) polar regions

**Methodology Validation**

Illustration of comparison with reference list events (black circles)

| PSH | ECLAT | match | PSTR | PSH | LOBE |
|---|---|---|---|---|---|
| C1 | 9812 | 6344 (65%) | 3069 (31%) | 6344 (65%) | <5% |
| C3 | 10420 | 6124 (59%) | 3928 (38%) | 6124 (59%) | <5% |

| PSTR | ECLAT | match | PSTR | PSH | LOBE |
|---|---|---|---|---|---|
| C1 | 10222 | 5985 (59%) | 5985 (59%) | 2346 (23%) | 1650 (16%) |
| C3 | 9406 | 6570 (70%) | 6570 (70%) | 1840 (20 %) | 748 (8%) |

| LOBE | ECLAT | match | PSTR | PSH | LOBE |
|---|---|---|---|---|---|
| C1 | 5287 | 1772 (33%) | 3154 (60 %) | <7 % | 1772 (33 %) |
| C3 | 4756 | 1045 (22%) | 3184 (67 %) | <11% | 1045 (22%) |

Comparison with ECLAT dataset (year 2003)

The comparisons with reference lists ( magnetopause: Trattner et al. (ESA); bow shocks: Kruparova et al. (2007); cusps: Pitout et al. (2006); plasmapauses: Darrouzet et al.(2013) ) show a very good agreement with the GRMB dataset (90%). The main limitation is the minimum resolution of the pre-generated plots (1-hour): some short back and forth boundary crossings are not detected.

Comparison with Bryant's plot

The comparison with the automatic list ECLAT (Boakes et al., 2006) is difficult due to the different time resolution (regions shorter than about 20 min  are not resolved in the GRMB dataset).

**Multi-SC and multi-years comparisons**

**Summary**

The GRMB dataset will provide a continuous coverage of the Cluster location of each of the four spacecraft for the whole mission duration.

15 labels are considered.

The dataset will be made available via  the Cluster Science Archive.

The visual selection is double-checked.

This dataset aims to support the scientific community not only by providing the location, but also to perform statistical studies. For example, the user can select the BSTR region to apply his/her own selection criteria for bow shock identification inside this transition region.

The realization of the GRMB dataset is funded by ESA Contract No. 4000139126/22/ES/CM  Geospace Region and Magnetospheric Boundary Identification using the Cluster Science Archive

---

## Author Response (AR1)

**Author Response to Reviewer comments**

First of all, we would like to thank both reviewers again for their time and effort in reviewing our paper and for their good and helpful comments. These have helped us to improve our paper significantly. In the following we response to all the comments raised by the reviewers.

This is a compilation of the comments we made in the public discussion. As we have already indicated where we have changed the revised manuscript, we have not changed the comments drastically.

**Specific Comments of reviewer 1:**

*The manuscript uses a lot of acronyms. The following comment is not a requirement for publication, but as a suggestion for readability and interpretation it is recommended to use fewer acronyms. Some examples of acronyms that aren't common in the field: MPC, MPTR, GRMB, or PSTR. They appear to all be defined in the text, so it isn't wrong, but the extensive use does make it difficult to follow at a number of places in the manuscript.*

> We tried to use fewer acronyms in our manuscript to make it easier to read. We mainly changed "MPC" to "crossing".

*81 – "sampled at a cadence of 60s for MPC identification.." clarify what this means in the text. Does this mean one data point is taken every 60s or data points in each 60s window are averaged or something else? Unclear.*

> We update the line in the manuscript to clarify. It now reads: "The magnetic field and plasma moments data are used in spin-averaged resolution with cadences of about 4 s during pre-processing and resampled by taking a data point every 60 s for MPC identification."

*91 – "data gaps of a few minutes.." need to be more specific in the text. What is "a few"? The goal of our work in peer-review research is reproducibility. There should enough detail should be provided so one could reproduce the results.*

> We agree with the reviewer about the reproducibility of the work. We clarified that we only interpolate data gaps smaller than 10 minutes.

*97 – "focusing on the energy flux density, ion density and magnetic field data for identification" More details needed in the text. Are specific thresholds are used, this needs to be listed. If not, the trends which the authors visually inspected for should be described. Rotations of B? Decrease or increase in density? Other?*

> As suggested by the reviewer, we have added more information in the paragraph to explain how we labelled the data. We did not use specific thresholds, but rather looked at jumps in density and magnetic field, as well as the appearance of the broader ion energy distribution around 3 keV associated with the magnetosheath, and the narrower ion energy distribution associated with the solar wind.

*104-108 - #3 and #4 claim "are statistically only observed outside the magnetosphere". This needs*

*more description. What does it mean? In any statistical study it's extremely hard (impossible?) to come up with strict rules which apply to every scenario and instrument mode across a dynamic system like the magnetosphere. The reviewer understands what the authors are generally trying to say, but the statement used needs more information to justify. Does "statistically" mean the authors have calculated how often it could be correct? What does it mean in this context? Text needs to be added to describe.*

We agree with the reviewer. In order to clarify this paragraph and to give a more detailed explanation of how we arrived at the thresholds, we have added a new Figure 1 to the manuscript, which shows that the data collected can be divided into two categories. The thresholds were then chosen to ensure that the filtered data points belonged to these categories, while also keeping literature values in mind.

*112 – need to define precision score metric in this manuscript. Pointing to another paper not sufficient for something critical to the reliability of this dataset.*

As requested by the reviewer, we have added a definition of the score used. It defines the fraction of correct predictions from the predictions that are labelled "inside the magnetosphere".

*134 – Define crossing probability in text. The phrase is used several times, but definition is missing. Likelihood of a data time series of a given length containing a crossing?*

Although the crossing probabilities were already partially defined in the text ("The crossing probability indicates how accurately the RFC can determine the labels of the data points around the MPC, thus providing a quantification of the ambiguity of the MPC"), we have expanded the paragraph to include the mathematical calculation of the crossing probability introduced in Grimmich et al. (2023) which is a weighted sum of prediction probabilities around the identified crossing.

*138 – Aberrated GSE – there are a few implementations currently being used in the community to define/implement this. Need to define in the manuscript how the study is doing this.*

We thank the reviewer for pointing this out, we were not aware of the multiple implementation and therefore omitted the definition in the first draft. We have now added a paragraph after Figure 2 explaining the coordinate transformation. We use an average aberration angle of 4.3 degree, resulting from the Earth's orbital velocity of 30 km/s around the Sun and an average solar wind velocity of 400 km/s, to rotate the normal geocentric solar ecliptic coordinate system about the z-axis into the aGSE system.

*154 – This definition is confusing and discouraged. Most other papers in this area define the stand-off distance as the distance of the magnetopause at the subsolar point. Here, the manuscript appears to use the "stand-off distance" to mean the radial distance of the boundary. This caused confusion throughout the reading for the reviewer.*

*The manuscript also flips the variable in equation (1). Shue uses Ro to describe the stand-off distance or the distance of the subsolar magnetopause (variable described as "r" here). The manuscript adopts (1) from Shue 98 but switches two variable names. This isn't technically wrong,*

*but it is very confusing for a reader. The reviewer recommends sticking to the common nomenclature used in the literature for ease of comprehension.*

80   We think that the reviewer may have misunderstood what we indented to say in the paragraph. We use the term 'stand-off distance' in its common definition as the radial distance to the subsolar point of the magnetosphere. The equation is a conversion of the SH98 formula from radial distance to stand-off distance, allowing us to map an observed radial position to a stand-off distance that is easily comparable to a modelled one. A similar approach has been taken in several previous papers (e.g. Plaschke et al., 2009; Staples et al.,

85   2020; Grimmich et al., 2023). For clarity, the explanation around equations (2) and (3) has been rephrased in the updated manuscript.

*161 – The magnetopause is often extremely difficult to identify near the exterior cusp. Both the sheath and exterior cusp have similar plasma populations in composition and spectra and can have similar flows. Past work using Cluster has often struggled to pick out a single point for the*

90   *boundary. The statement about a dimple somewhat over simplifies this region.*

We agree with the reviewer that the cusp is a plasma regime that is difficult to separate from the magnetosheath. This is the reason why we have tried to discuss in this paragraph the problems arising from the use of the SH98 model when dealing with a mangetopause crossing near or at the cusp. Having said that, we have rephrased the paragraph to better

95   illustrate the complicate nature of the cusp.

*175 – Use of the GRMB is described as important to this manuscript validation. Right now very little description of this product is given in the manuscript and a reference to an EGU abstract is provided. This doesn't allow a reader to learn about the details of the methodology. This is an issue that needs addressing. One possibility is to delay this manuscript until a more complete*

100   *description of GRMB is available. A second possibility could be to provide more details in the current manuscript to allow the reader to understand how it works and its performance. The authors are free to come up with other solutions as well, but something needs to be done to address. Without a detailed description of GRMB and its validation the reader doesn't know the fidelity of that data set. If the GRMB dataset is poorly defined or uses very large time averages,*

105   *one may not expect agreement between the two studies to actually validate the work presented here.*

*The reviewer does credit the authors for including the step of validation. It's the right thing to do and common-place in many fields, but is sadly often left out in areas of space physics research.*

We agree with the reviewer's comment. The delivery of the GRMB dataset to the CSA has

110   currently started and the first data products should be available soon: the methodology has been already validated by ESA in the frame of the   contract No. 4000139126/22/ES/CM (the corresponding documents are not made public by ESA). We believe that the validation of the methodology by ESA is allowing us not to wait for the publication or the full dataset description, which is under preparation.

115   We opt for the second option proposed by the referee, namely "provide more details in the

current manuscript to allow the reader to understand how it works and its performance" by adding a extended paragraph in the manuscript describing the magnetopause identification in GRMB.

We also attached to this reply two posters presented at AGU-2023 and EGU-2024, posters describing the GRMB methodology.

*Fig 5 caption – What is the "second bin"? Needs better description.*

The normalisation in Figure 5A was done with the bin between 5 and 10 degrees deviation between the estimated and modelled MP normal. We have rewritten the caption for clarity.

*Line ~247 - Section 4. The reviewer is unable to find a definition used for high latitude. Please clarify in the manuscript. Where data filtered to only include a subset with an latitude greater than a threshold?*

Yes, the data were filtered using a latitude threshold. The thresholds used are also already specified in the manuscript in section 3: "For the MPCs in the high latitude regions ($|\theta| > 30°$), on the night side ($|\phi| > 90°$) and also slightly on the flanks ($|\phi| > 30°$), …". Nevertheless, we repeat our definition for the high-latitude region in section 4.

*311 – "Only for the cusp MCPs…" The reviewer isn't able to identify a definition used for the cusp. Does the study use some criteria to identify the cusp or is this purely based on a model and spatial location? Need to clarify in text.*

As stated in the manuscript (last paragraph of section 3), we assume that the MPCs found in the area where the cusp is most likely to be located are cusp encounters ("…MPCs occur in the area where the cusp is most likely to be located. We define this as the area where $|MLAT| > 70°$ and $|MLAT| \leq 85°$ and $MLT \geq 10$ and $MLT \leq 14$ holds. A total of 593 MPCs (383 well-defined) fully meet the criteria and fall 170 between the MLAT and MLT areas, most likely related to the cusp location."). In section 4 we state: "The first subset gathers the crossings founds where the cusp is most likely to be located (28 MPCs).", which is a reference to the previous definition. To make this a little clearer in the text, the sentence in section 4 has been expanded.

*354 – Text needs to be added to describe how OMNI dataset is used. Is a time step from OMNI grabbed to match the time step from the spacecraft or is some other propagation method included?*

In Section 5, the OMNI data associated with an MPC is treated in the same way as the OMNI data used to calculate the distance (described in Section 2). This means that an average is taken in an 8-minute time interval of the OMNI data prior to the identified crossing time. We made it clear at the beginning of section 5 that this is how we treat the data.

*357 – This is a bit of a strange statement. Generally a study stands on its on as an independent work. Here, rather than making an independent assessment of what's important, the work asserts that since a previous study found some parameters to be unimportant, they wont be considered.*

*The reviewer doesn't present any required changes with this comment, but this is noted as a comment to the authors to consider.*

> We thank the reviewer for pointing this out. In order to make our manuscript more independent from the previous work of Grimmich et al., we decided to include the other parameters and thus update Fig. 8, added a Fig. 9 and rephrased some of the following descriptions in the revised manuscript.

*Paragraph starting at 366 – Given the error bars the reviewer does not feel all the trends stated have been shown to be statistically significant. The work needs to apply a statistical test to show. A simple students-t test or other statistical metric could show a probability that the different distributions are real or simple due to statistical variations.*

> We agree that some visible differences may not be statistically significant, so we applied statistical test to our subset of data, as suggested by the reviewer. Since the solar wind parameter distributions are not normally distributed, we applied the Mann-Whitney U test, a t-test for non-normally distributed data. We have summarised the results of the test in Table 1 of the manuscript and briefly discussed the results in the surrounding text. The bold values indicate significant deviations that are not due to chance. Therefore, most of our findings can be considered valid.

*Figure 7 – The definition of the error bars needs to be described.*

> We are not sure whether the reviewer means that the description should be added to the figure caption or whether the explanation in the text needs to be more substantial. Nevertheless, we have expanded the description of the error estimate in the manuscript text by adding equation (7): bin error = 2/(data set size * bin size of solar wind reference).

*397 – Open/closed – More description needed to justify these claims. Measuring if a region is on open or closed field lines if often determined by looking at pitch angles of particles and sometimes species. Here the authors are using the magnetic field in the magnetosheath to make assertions on open/closed. Types of things one would need to know - How is the region in the sheath identified? How long of a sheath region is used? Is there time averaging? How does one know the sampling is adjacent to the boundary versus further into the sheath?*

*This statement about open/closed boundary is repeated in the conclusions. It needs to be justified by better describing the experiment in the text.*

> In our revised manuscript, we have tried to go into more detail on this issue by answering some of the questions raised by the reviewer. We added a brief explanation how the magnetosheath data was gathered. From the time series used for identification, we took the vector at the timestamp just before or just after the crossing, depending on where the RFC label would indicate the "outside the magnetosphere" label. This is, of course, only a very rough estimate of the magnetosheath field outside the MP. However, this analysis was just a quick test to see if we could glean any information by comparing the MP normals and the magnetosheath field.

*413-414 – "..Although not shown in much detail in this study, the results from the equatorial plane*

*of Cluster agree well with the results from Grimmich et al. (2023a)." There are several claims throughout the manuscript which cite closely this previous work by the author. As an independent study this manuscript needs to be able to live on its own and justify its own claims. This particular statement needs more description here or removal.*

We agree that our paper should stand on its own. We have tried to minimise the references to the previous paper and explain more about what we found in this study. During this process, the highlighted statement was removed.

*516 - #4 – this needs to be more specific. The reviewer things the authors are intending to say the dipole tilt angle has a significant effect on the magnetopause position? It would also be helpful to quantify with numbers.*

We agree with the reviewer and have added some quantification to the description of Fig. 10. Furthermore, an estimate of 2 Re of deviations from the MP model locations due to the tilt angle has been included in the expanded point #4 in the section "Conclusions".

*517 – "Together with the equatorial MPCs from the THEMIS dataset (Grimmich et al., 2023b) we now studied…" Similar comment to before – the study needs to stand on its own. This seems to be a conclusion for both works. Text needs to be modified to cover the claims and justifications documented here in this study.*

We have rephrased the sentence in the conclusions to further separate our findings from the previous study, as suggested by the reviewer. It now reads: "Since the previous study by Grimmich et al. (2023a) investigated large deviations from the modelled MP position in the equatorial region, and this study now extends this investigation to the high latitudes, we can begin to formulate a more global behaviour of the MP response to solar wind influences beyond the dynamic pressure."

*It would be helpful to have a figure with an actual magnetopause crossing identified by the algorithm shown, particularly during a complicated crossing with a rapidly moving boundary and mixed plasmas. Currently there are no figures with such data are shown. This is a recommendation and not a requirement for publication or correctness.*

We thank the reviewer for the suggestion and have added the two time series plots of various magnetopause crossings identified to our appendix.

**Major concerns of Jonas Suni:**

*Considering the importance of Cluster being able to observe at high latitudes, there should be a brief description of the Cluster orbits both in the abstract and early in the introduction.*

We agree with the reviewer and thank him for this suggestion. We have included a brief description of the orbits in the abstract and the introduction to our manuscript.

*While understandable, the decision not to describe the MPC identification method in Grimmich+2023b is, in this reviewer's opinion, an issue. For instance, the significance of the "precision score" of 0.998 on line 113 is completely unclear, as is the "crossing probability" on line*

*115. It is also unclear what the average solar wind velocity mentioned on line 139 is used for, is it for this method or something else? Finally, given the 4 criteria for finding MPCs on lines 99-109, it is unclear why an ML-based approach is even needed at all. This reviewer recommends adding a short description of what input the MPC identification method takes, what it outputs, and what the precision score and crossing probability mean.*

To clarify our identification method, as recommended by the reviewer, we have added additional description and information in our section 2. We have tried to address all of the reviewer's concerns. For example, we have expanded the definitions of precision score and crossing probability in the text, while also clarifying that we only used the thresholds to balance our training dataset and that they would not be suitable for identifying regions near the boundary of the magnetosphere. A fully threshold-based approach could also have been applied to the Cluster data. However, as a machine learning approach is more adaptable and we had already identified a very suitable algorithm in our first study, we decided to use it again, although some retraining was necessary. Regarding the average solar wind velocity mentioned by the reviewer, this velocity was used to calculate the aberration caused by the Earth's orbit around the Sun. We have also extended this paragraph with more details.

*The topological configuration of the magnetopause is important for understanding how a closed surface model like SH98 differs from the necessarily open field at the cusps. This is alluded to in the text, but a short description in the introduction of the topology of the actual magnetopause would make the following results easier to understand.*

We thank the reviewer for his suggestion and include a brief description of this topic at the beginning of our introduction.

*Throughout the manuscript, the comparisons between the Shue model and the observed MPCs are done in terms of the magnetopause standoff distance Ro. Given that x and r are known for the observed MPCs, it is unclear why these are transformed into magnetopause standoff dsitnace rather than calculating the x and r from the Shue model and comparing those to the observed values. A brief explanation of why this is done would be very helpful.*

The reason for converting the location to the distance is that a direct comparison using r or x would introduce a significant bias, especially further away from the sub-solar point. Due to the curvature of the MP surface, the difference vector between the radial position of the spacecraft and the modelled MP position is not necessarily perpendicular to the surface, resulting in apparently large deviations. This is in some way avoided by comparing the stand-off distances. Therefore, in order to better compare observations and predictions, and also to have a similar method between this study and the previous study by Grimmich et al. (2023), we use the transformation. Furthermore, this has already been done in other studies (e.g. Plaschke et al. (2009a, 2009b); Staples et al. (2020); Grimmich et al. (2023a)). To emphasise this in our manuscript, we have added a paragraph explaining our decision.

*The manuscript would benefit greatly from additional figures that show a timeseries example of a magnetopause crossing as well as schematics that depict outbound and inbound magnetopause*

*crossings, or in general just a magnetopause crossing.*

We agree, and since the other reviewer also suggested this, we have added two figures to the appendix of our paper showing time series data labelled by the machine learning algorithm from which we would identify crossings. We show an inbound and an outbound crossing with multiple passages through the boundary layer.

*On lines 200-202, the authors refer to by-eye selection vs. automatic classification. Presumably this means that the GRMB dataset was classified by eye and the MPC dataset in this study automatically, but this is not clear. Which dataset was collected how should be stated explicitly.*

We agree with the reviewer that it should be emphasised how the two datasets were constructed. We have added the description of the classification method to the relevant paragraph in our updated manuscript.

*On lines 281-282, the sentences starting with "Nevertheless," and "Thus," appear to contradict each other. It is unclear when all MPCs are considered and when just high latitude MPCs are considered. Please revise.*

We agree with the reviewer that the sentences are written in a cantadictory way. We have therefore revised the statement to clarify that we are always talking about high latitude MPCs and only show the distribution for all MPCs in the plots for reference.

**Minor concerns of Jonas Suni:**

*On line 3, the sentence would be clearer if rephrased as "The magnetopause is the boundary between the interplanetary magnetic field and the terrestrial magnetic field." The next sentence could also start with "It is in influenced..." rather than "this magnetopause is influenced...".*

We thank the reviewer for his suggestion and rephrased the sentences accordingly.

*The sentence starting with "This magnetopause is infleunced..." on line 3 has a redundant clause. The "influenced by dynamic changes in the solar wind" could be rephrased away in favour of just the following description of what it means.*

We thank the reviewer for his suggestion and rephrased this sentence as well.

*Line 37: Presumably the authors mean that Dorville+2014 are referring to the MP during southward IMF.*

Yes, the citation from Dorville et al. refers to the description of the MP under southward IMF conditions. We have rephrased the sentence to make this clear.

*Line 48: "IMF strength and orientation" is a bit ambiguous given the full sentence, "IMF strength and IMF orientation" would be more clear. Also, "dynamical pressure" should be "dynamic pressure" consistently throughout the manuscript.*

We thank the reviewer for his suggestion and have changed the manuscript accordingly.

*The definition of R depends on the sign of y but not z. How does this affect the estimation of the importance of the dipole tilt angle (where the sign of z is important)?*

The definition of R has no effect on the dipole tilt angle estimates, as we only used R for the panel (a) of Fig. 2. For the other calculations, including the tilt angle estimates, we always used all three coordinates. To avoid further misunderstandings, we note this fact in the definition of R.

*Line 145: How is the time delay from the bow shock to the MP and terminator estimated?*

For the 8 min delay mentioned in the text we consider the following: To estimate this time delay, we assumed typical distances between the bow shock and the MP of 3 to 4 RE, typical distance form the MP to the terminator of 10 RE and typical flow velocities in the subsolar (flank) magnetoseath of 100 km/s (300 km/s). Using these numbers to calculate how long it would take a plasma element to travel from the BS to the MP and then to the terminator gives the 8 minutes we considered as the time delay. We have also included this description of the estimate in the manuscript.

*Throughout the manuscript, there are many abbreviations which occasionally make the text difficult to read. Could some of these be replaced by regular words? For instance, after the Cluster instrument names are introduced, could "FGM data" be rephrased as "magnetic field data", "CIS data" as "moments/plasma data", MPC as "crossing" (when it is clear that is what is being talked about), etc?*

We followed the reviewer's suggestion and tried to minimise the use of acronyms in the updated text.

*The GRMB labels should be defined in the text as well, not just in a figure caption.*

We agree and added short definitions of the labels used in the text.

*In figures 3-8, it is not always clear why the histograms are normalised to the values of certain bins. This should be explained. Additionally, the labels of the vertical axes vary between "norm. # MPCs/hour", "norm. MPCs count" and "norm. MPC occurrence". Is there a reason why these are worded differently, and do they actually mean the same thing? This should either be explained or the vertical axes changed to the same quantity and wording.*

The labels "norm. MPCs count" and "norm. MPC occurrence" should have been the same and we fixed this in the revised manuscript. The label "norm. # MPCs/hour" is different because the histograms in Fig. 4 and 5 are normalised differently. In order to explain this better, we have added further explanations to the manuscript, referring to the relevant figures. Furthermore, we chose the bin on which to normalise according to expected to theoretically known values, e.g. we expect the angular deviations between MP normals to be only 5 degrees, or typical MP velocities to be around 50 km/s.

*On line 274, it is stated that the timing method is deemed reliable if the lowest cross-correlation coefficient is greater than 0.65. How was this value chosen? E.g. Eastwood+2005 (https://doi.org/10.1029/2004JA010617) use 0.8.*

The value of 0.65 was chosen arbitrarily. We did not want to choose a value that was too high, which might have resulted in more events being discarded, leaving our study with

345        too few events to analyse properly.

*On line 308, the authors state that the median inward MP motion is 73.6 km/s. Then, the statement on line 306 that MP velocities between 0 and 75 km/s are "most common" is just barely true, as half of the velocities are above 73.6 km/s. This does not necessarily require revision, but it could lead to misunderstandings if only one of these numbers is referenced in future studies.*

350        We agree with the reviewer on this point and have changed the phrase "most common" to "... many of the crossing observations were made at low velocities between 0 and 75 km/s ...".

*The description of the linear regression referred to on line 328 would benefit from and equation or figure.*

355        We have added the equation of fit to the line referred to above.

*Throughout the paper there are several very long sentences that are confusing due to their length, for instance the sentences on line 348 and 480.*

        We agree with the reviewer and have tried to rephrase complicated sentences in the revised manuscript.

360 *The sentence "we compute the quotient of the distributions associated with MPCs with the solar wind occurrence rate distributions" on line 360 is difficult to understand, consider rephrasing.*

        We have reworded the sentence to clarify our normalisation. It now reads: "... we normalized the count rates per bin. The normalisation is done by dividing the parameter distributions associated with the crossings by the natural solar wind distribution in the

365        years 2001 to 2020."

*What is the statement "since we find on average 2 MPCs per h in all 1 h intervals" on line 363 based on? Is this a result based on a figure? If so, please refer back to the figure in question.*

        This statement is based directly on our database by counting how many MPCs are found in the 1 hour intervals we examined and taking the average of the resulting distribution. The s

370        entence in question has been rephrased slightly in the manuscript.

*On line 393, the "MPC identification method ... was applied to the Cluster dataset after some retraining" is vague, consider elaborating on what retraining was done.*

        We agree with the reviewer and added more to the beginning of our discussion.

*Consider elaborating on the following:*

375 *On line 399, "... some sort of overfitting" is vague, consider elaborating on this as well.*

        A brief explanation of this statement has been included in the revised manuscript.

*On line 403, compared to what are the "MP stand-off distances" low?*

        The stand-off distance is low compared to the usual value of 10 RE. This is explained in the revised manuscript.

380 *On line 407, what is the "2 RE" an estimate of?*

> The value 2 RE is the average depth of the cusp and the bias steming from the indentation on the magnetopause surface. We expanded the manuscript to emphasise this.

*On line 418, compared to what are the "MPCs in the Cluster dataset" drastically more compressed?*

385
> The reviewer misunderstood the sentence. We wanted to say that we have more compressed MPCs than expanded MPCs. To avoid further misunderstanding, we rephrased the statement in the revised manuscript.

*On line 503, in which cases was "the dataset used in this research" very limited due to the applied selection and filter criteria?*

390
> This statement refers mainly to the expanded MPCs, which are very few compared to the compressed MPCs at high latitudes. We have tried to make this clearer in the revised manuscript.

*On line 516, on what quantity is the "dipole tilt angle influence significant" at high latitudes?*

> The influence of the dipole tilt is on the MP position. We have rephrased this point in the
395 conclusion for clarity, and now also give an indication of how much deviation the dipole tilt can cause.

---

## Referee Report (RR1)

**Review egusphere-2024-1087**

The authors have addressed all of the reviewer's comments in a satisfactory manner in the author response. The authors have incorporated all of the requested changes into the revised manuscript and elaborated sufficiently on all unclear parts of the text. In the reviewer's opinion, this has helped to make the presentation of a very important study more clear, more robust, and more accessible to the prospective reader. The reviewer recommends **publication subject to technical corrections,** with the requested corrections listed at the end.

I thank the authors for engaging in discussion about the study. To re-iterate my original review, I think this paper will be of significant importance in understanding the behaviour of the magnetopause.

**Technical corrections:**

- On line 164, the sentence part "small but bias" appears to be missing a word.

- On line 573, the sentence part "more southward IMF conditions with plasma beta < 1 lower ion density and faster solar wind velocities" appears to be missing a comma.

Best regards,

Jonas Suni